# RECOVER: Reliable Detection of Unauthorized Data Usage in Text-to-Image Diffusion Models via Inversion Robustness

**Yanhao Wei** [* 1] **Xiaokang Zhao** [* 1] **Boheng Li** [* 2] **Yang Zhang** [1] **Run Wang**[✉ 1]

## Abstract

Text-to-Image diffusion models have achieved remarkable success in image generation and are increasingly fine-tuned for personalized use cases. However, many personalized models may incorporate unauthorized data during the fine-tuning process, raising growing concerns about potential copyright infringements. Existing methods either require intrusive modifications to the images to be protected, which not only fail to safeguard previously released images but may also degrade image quality, or rely on the availability of the pre-fine-tuned model, thereby limiting their applicability. To bridge this gap, in this paper, we propose the first non-intrusive copyright authentication framework without pre-fine-tuned model. We reveal that if a model is fine-tuned on a specific image, it learns the denoising trajectory of that image across varying noise levels, allowing it to stably reconstruct the image even under noise perturbations. Motivated by this insight, we propose **R**eliable d**E**te**C**tion **O**f unauthorized data usage via in**VE**rsion **R**obustness (RECOVER), an effective non-intrusive detection method without pre-fine-tuned model. Unlike existing methods that rely on external watermarks or discrepancies between the suspect and pre-fine-tuned models, RECOVER directly leverages the robustness observed during the inversion–reconstruction process of the suspect model to determine whether an image was used for fine-tuning. Extensive experiments demonstrate that RECOVER is effective across a wide range of scenarios, consistently outperforming existing methods. Our code is publicly available here.

*Equal contribution [1]Key Laboratory of Aerospace Information Security and Trusted Computing, Ministry of Education, School of Cyber Science and Engineering, Wuhan University [2]Nanyang Technological University. Correspondence to: Run Wang <wangrun@whu.edu.cn>.

*Proceedings of the 43$^{rd}$ International Conference on Machine Learning*, Seoul, South Korea. PMLR 306, 2026. Copyright 2026 by the author(s).

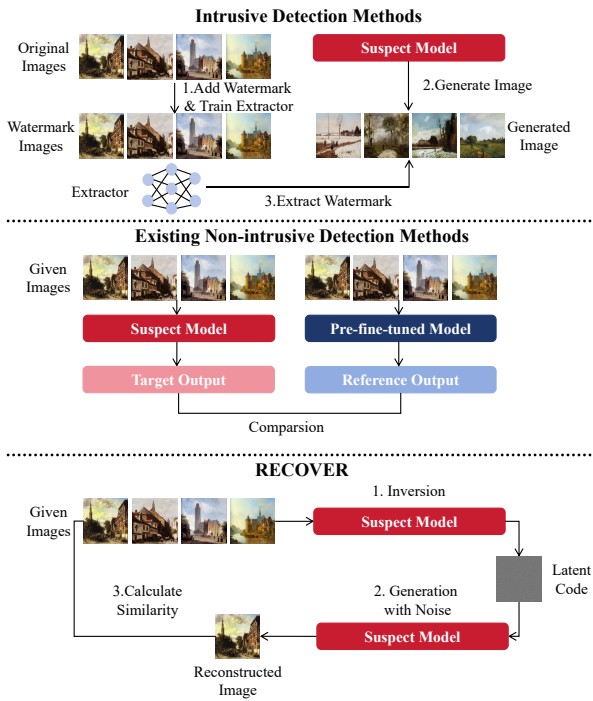

*Figure 1.* **Illustration of the pipelines of RECOVER and existing methods. Top:** Intrusive detection methods require embedding watermarks into images in advance and verify their presence in outputs from the suspect model. **Middle:** Existing non-intrusive detection methods rely on comparing outputs between the suspect model and the pre-fine-tuned model. **Bottom:** RECOVER performs authentication by comparing the inversion–reconstruction consistency of images using only the suspect model.

## 1. Introduction

Diffusion models (Ho et al., 2020; Song et al., 2021a;b) have revolutionized generative AI technology, enabling the synthesis of images from randomly sampled latent codes. In particular, large-scale pre-trained text-to-image (T2I) diffusion models (Rombach et al., 2022; RunwayML, 2022; AI, 2022a; Ye et al., 2024), such as Stable Diffusion (Rombach et al., 2022), can generate high-quality and diverse images conditioned on user-provided prompts. In addition to their image generation capabilities, another key feature of T2I diffusion models is their ability to perform personalized

generation, enabling them to learn specific objects or styles through few-shot fine-tuning. Techniques like LoRA (Hu et al., 2022) and DreamBooth (Ruiz et al., 2023) facilitate this process, enabling models to efficiently generate content that is not only increasingly realistic, but also closely aligned with the intended thematic elements.

However, the remarkable success of these advanced few-shot fine-tuning strategies has raised concerns about unauthorized data usage, as a fine-tuned T2I diffusion model can easily generate images that closely replicate the style of the fine-tuning dataset. For instance, artists may fear their artwork without permission to fine-tune diffusion models, thereby generating images with a similar artistic style for profit, which could lead to severe financial damage (Jiang et al., 2023) and copyright violations (Shan et al., 2023). Although the data owner may observe that a model generates images resembling the style of their dataset, they often lack sufficient evidence to prove that the model was fine-tuned using their data, making it challenging to pursue legal action.

To mitigate these risks, a growing number of efforts (Sablay-rolles et al., 2020; Li et al., 2022; Wang et al., 2024; Cui et al., 2023; Li et al., 2025) attempt to trace data usage through intrusive methods. As illustrated in Fig. 1 (Top), they require the addition of an external watermark to the image before release, thereby enabling traceable detection of unauthorized data usage. However, these methods require embedding watermarks before the dataset is released, rendering them ineffective for datasets that have already been made public. Moreover, these methods often degrade image quality due to the need for an additional watermark. For images that emphasize details (*e.g.*, artworks) these methods often present significant drawbacks.

Consequently, a line of recent research (Wu et al., 2024; 2025) has explored non-intrusive detection methods. As illustrated in Fig. 1 (Middle), these approaches compare the suspicious model with its pre-fine-tuned counterpart and leverage their differences to obtain strong signals of potential infringement. For instance, the state-of-the-art method CGI-DM (Wu et al., 2024) reconstructs images by exploiting the gradient discrepancies between the two models, while FineXtract (Wu et al., 2025) extracts fine-tuning data by leveraging differences in their predicted noise. However, these methods fail when the pre-fine-tuned model is unavailable or has been maliciously claimed, thereby limiting their applicability. In real-world scenarios, a malicious adversary is unlikely to provide the genuine pre-fine-tuned model.

To bridge this gap, in this paper, we attempt to propose the first effective non-intrusive detection method that does not require access to the pre-fine-tuned model. First, we reveal that, because the model learns to denoise fine-tuning images across various noise levels, these images exhibit a more

stable inversion–reconstruction ability under noise perturbations. We refer to this property as *inversion robustness*, and its visualization is shown in Fig. 2. We further demonstrate the generality of this phenomenon through comprehensive empirical validation and preliminary theoretical analysis in Sec. 3.3.

Motivated by this insight, we propose RECOVER (**R**eliable **DE**te**C**tion **O**f Unauthorized Data Usage via In**VE**rsion **R**obustness), the first effective non-intrusive detection method without pre-fine-tuned model. As illustrated in Fig. 1 (Bottom), RECOVER only requires evaluating the inversion–reconstruction ability of images on the suspect model as a strong indicator for identifying potential unauthorized data usage, without the need for intrusive modifications or access to the pre-fine-tuned model.

Extensive experiments were conducted on 8 mainstream text-to-image diffusion models and 2 benchmark datasets with 3 popular few-shot fine-tuning methods. The results show that our method is highly effective and significantly outperforms 4 state-of-the-art baselines. It exhibits high transferability across various fine-tuning algorithms, fine-tuning parameters, and base models. We further evaluate the robustness of our approach in several challenging scenarios (*e.g.*, limited data ratios).

Our contributions are summarized as follows:

- We reveal and demonstrate that fine-tuning images exhibit stronger inversion–reconstruction ability under noise perturbations, as the model learns to denoise these images across various noise levels.

- Motivated by this insight, We introduce RECOVER, the first effective non-intrusive detection method against unauthorized data usage that operates without requiring access to the pre-fine-tuned model.

- We systematically validate the effectiveness of RECOVER on various datasets, models, and few-shot fine-tuning algorithms. Moreover, we validate its robustness under challenging settings.

## 2. Related Work

### 2.1. Diffusion Model & Few-shot Fine-tuning

**Diffusion Models.** Diffusion models (Ho et al., 2020; Song et al., 2021a;b) are generative models that generate images by gradually denoising a latent code sampled from a prior distribution. These models involve a forward process and a backward process. In the forward process, the input image $x_0$ is perturbed into a noise image $x_t$ following $x_t = \sqrt{\alpha_t} x_0 + \sqrt{1 - \alpha_t} \epsilon$, where $\alpha_t$ denotes a set of predefined parameters for each timestep $t$, and $\epsilon \sim \mathcal{N}(0, \mathbf{I})$. In

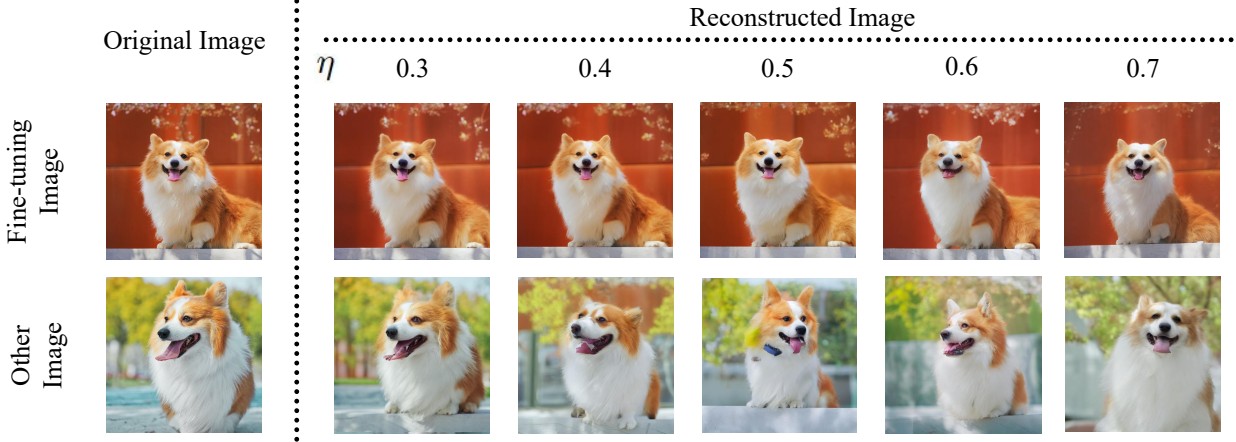

*Figure 2.* Visualization of *inversion robustness*. Images are first inverted via DDIM inversion (Song et al., 2021a) and then reconstructed under different stochasticity ratios $\eta$ (Definition 3.1). It can be observed that fine-tuning set images maintain strong reconstruction ability even at higher noise levels, whereas other images are difficult to reconstruct. Additional visualizations can be found in App. H.1

contrast, the backward process employs a noise prediction model $\epsilon_\theta$ to estimate the noise $\epsilon$ added to noise image $x_t$. Consequently, the training objective of the diffusion model is to minimize the discrepancy between the predicted noise and the actual noise, which is defined as the diffusion loss $\mathcal{L}_{\text{DM}} = \mathbb{E}_{x_0, t, \epsilon \in \mathcal{N}(0, \mathbf{I})} \|\epsilon - \epsilon_\theta(\sqrt{\alpha_t}x_0 + \sqrt{1 - \alpha_t}\epsilon, t)\|^2$.

**Text-to-Image Diffusion Models & Few-shot Fine-tuning.** T2I diffusion models (Rombach et al., 2022; RunwayML, 2022; AI, 2022a; Ye et al., 2024) have achieved unprecedented success in the field of text-to-image generation. Among them, Stable Diffusion (Rombach et al., 2022; RunwayML, 2022; AI, 2022a), a model based on the latent diffusion model architecture (Rombach et al., 2022), is currently one of the most influential text-to-image diffusion models. This model performs the diffusion process within a latent space generated by a pre-trained autoencoder, represented as $z = \mathcal{E}(x)$, where $\mathcal{E}$ denotes the encoder. This allows the model to leverage the highly compressed semantic features and visual patterns learned by the encoder. The pre-trained T2I diffusion model, known as the base model, is capable of generating high-quality images based on prompts. Few-shot fine-tuning (Ruiz et al., 2023; Hu et al., 2022) facilitates personalized generation by adapting a base model to incorporate new concepts from a small fine-tuning dataset, thereby enabling the creation of diverse images with specific themes or styles.

### 2.2. Defending against Unauthorized Data Usage

**Intrusive Detection Methods.** These methods mainly includes adversarial-based defenses (Shan et al., 2023; Liang et al., 2023; Van Le et al., 2023) and dataset watermarking (Sablayrolles et al., 2020; Li et al., 2022; Wang et al., 2024; Cui et al., 2023; Li et al., 2025). These methods require the addition of perturbations or watermarks before image

release to either prevent the target model from learning specific concepts or enable traceable detection. However, both adversarial attacks and dataset watermarking require intrusive modifications to the images, which makes them ineffective for protecting copyrighted images that have already been made public. On the other hand, these methods may degrade image quality due to the need for an additional perturbation. Moreover, the emergence of adversarial perturbation removal attacks (Cao et al., 2023; Hönig et al., 2025) and watermark removal attacks (Datta et al., 2024) has reduced the effectiveness of these methods.

**Non-intrusive Detection Methods.** Existing non-intrusive detection methods rely on the discrepancies between the suspect model and its pre-fine-tuned counterpart to construct a strong signal for infringement accusation. For instance, CGI-DM (Wu et al., 2024) reconstructs corrupted images using gradient discrepancies and then measures the similarity between the reconstructed and original images as evidence of infringement. Similarly, FineXtract (Wu et al., 2025) exploits noise discrepancies to extract fine-tuning images, using the similarity between the extracted and original images as the infringement signal. However, these methods all rely on the outputs of the pre-fine-tuned model as a reference, which makes them ineffective when the pre-fine-tuned model is unavailable or has been maliciously claimed. A malicious fine-tuner does not provide access to the genuine pre-fine-tuned model in real-world scenarios. We provide the corresponding quantitative results in the App. A.1 for scenarios where the pre-fine-tuned model is maliciously claimed. Originally designed as privacy-invasive attacks, membership inference (Shokri et al., 2017; Carlini et al., 2022; Duan et al., 2023; Dubiński et al., 2024; Pang et al., 2024; Kong et al., 2024; Zhai et al., 2024) can serve as potential non-intrusive detection approaches. We discuss

the limitations of these methods when applied to copyright protection in the App. A.2.

## 3. Method

### 3.1. Threat Model

We focus on serving as a defender to determine whether a T2I diffusion model has been fine-tuned on a given image.

**Infringer's Capabilities & Goals.** The infringer aims to fine-tune a T2I diffusion model without authorization, enabling it to generate high-quality mimicry images. For certain purposes (*e.g.*, profit), they publicly release fine-tuned model $\theta$ and the fine-tuned text condition $c_{ft}$ that can generate images with specific objects or styles on fine-tuning platforms (Civitai, 2024; Replicate, 2024; Scenario, 2024). They have full control over the fine-tuning process and potentially apply certain modifications to the collected images.

**Defender's Capabilities & Goals.** For a given model $\theta$, a fine-tuned text condition $c_{ft}$, and a given image $x$, the defender's goal is to determine whether the image $x$ has been used to fine-tune the model $\theta$. When a legal prosecution is conducted, the defender is assumed to have white-box or gray-box access to the accused model $\theta$.

In the App. G, we include a more detailed discussion regarding the potential adaptation of our method to flexible black-box settings.

### 3.2. Preliminary

**DDIM Inversion.** DDIM inversion (Song et al., 2021a) is a technique that reverses the sampling process to derive the latent codes of the original images with text condition. Given the image $x_0$ and text condition $c$, it enables the estimation of latent code $z_T$ by iteratively applying the following equation:

$$z_{t_I} = \sqrt{\alpha_{t_I}} \cdot \hat{z}_0 + \sqrt{1 - \alpha_{t_I}} \cdot \epsilon_\theta(z_{t_{I-1}}, c) \quad (1)$$

where, $\hat{z}_0 = \frac{z_{t_{I-1}} - \sqrt{1-\alpha_{t_{I-1}}} \, \epsilon_\theta(z_{t_{I-1}}, c)}{\sqrt{\alpha_{t_{I-1}}}}$, $t_I$ and $t_{I-1}$ represent the time steps preset by the DDIM scheduler, and $z_{t_I}$ is iteratively obtained from $z_0 = \mathcal{E}(x_0)$. For convenience, we define a notation $\mathbf{Inv}(x_0, c)$ to represent the entire inversion process (*i.e.*, iterative application of Eq. (1)).

**DDIM Generation.** DDIM generation (Song et al., 2021a) is a sampling method for diffusion models that generates images based on text condition and latent code. Given the latent code $z_T$ and text condition $c$, it generates the image $x$ by iteratively applying the following equation:

$$z_{t_{I-1}} = \sqrt{\alpha_{t_{I-1}}} \cdot \hat{z}_0 + \sqrt{1 - \alpha_{t_{I-1}} - \sigma_{t_I}^2} \cdot \epsilon_\theta(z_{t_I}, c) + \sigma_{t_I} \epsilon_{t_I} \quad (2)$$

where, $\hat{z}_0 = \frac{z_{t_I} - \sqrt{1-\alpha_{t_I}} \, \epsilon_\theta(z_{t_I}, c)}{\sqrt{\alpha_{t_I}}}$, $\epsilon_{t_I} \sim \mathcal{N}(0, \mathbf{I})$ is standard Gaussian noise independent of $z_{t_I}$ and $\sigma_{t_I}$ is a coefficient that controls the level of randomness. When $\sigma_{t_I} = 0$, the process becomes deterministic DDIM generation, whereas when $\sigma_{t_I}$ satisfies the following equation:

$$\sigma_{t_I} = \sqrt{(1 - \alpha_{t_{I-1}})/(1 - \alpha_{t_I})} \cdot \sqrt{1 - \alpha_{t_I}/\alpha_{t_{I-1}}} \quad (3)$$

the forward process becomes Markovian and the generative process becomes a stochastic DDPM sampling (Ho et al., 2020). Through iterative application of applying Eq. (2), we first obtain $z_0$, which is then decoded into the image $x$ via the decoder $\mathcal{D}$, such that $x = \mathcal{D}(z_0)$.

### 3.3. Inversion Robustness

To further control the degree of randomness in the diffusion model's generation process, we provide a formal definition of the stochasticity ratio in the generative process.

**Definition 3.1** (Stochasticity Ratio). In a generative process, given the randomness coefficient $\sigma_t$, the diffusion coefficient $\alpha_t$, and the DDIM scheduler's preset time sequence $T$ (*i.e.*, $t_0 \ldots t_{I-1}, t_I \ldots$), we define the **stochasticity ratio** at any time step $t_I$ as follows:

$$\eta = \frac{\sigma_{t_I}}{\sqrt{(1 - \alpha_{t_{I-1}})/(1 - \alpha_{t_I})} \cdot \sqrt{1 - \alpha_{t_I}/\alpha_{t_{I-1}}}} \quad (4)$$

where $\eta$ denotes stochasticity ratio. Based on this definition, we can more precisely control the stochasticity of the generation process by adjusting the stochasticity ratio. For convenience, we also define a notation $\mathbf{Gen}(z_T, c, \eta)$ to represent the entire generative process. It is worth mentioning that when $\eta = 1$, the generative process is entirely stochastic, losing any connection to the inversion process. In this case, the model simply performs a standard random generation rather than reconstructing a specific image.

Since the objective of the diffusion loss $\mathcal{L}_{\text{DM}}$ is to optimize the model's ability to denoise images under various levels of stochastic noise, the model tends to reconstruct fine-tuning images more effectively under noise perturbations. For each denoising step in the DDIM generation process (according to Eq. (2)), fine-tuning images enable the model to more accurately predict $\hat{z}_0$, while other images struggle to do so due to the introduced noise. As the denoising steps accumulate, this prediction error is progressively amplified. We refer to this insight as inversion robustness, for which we first provide a formal definition.

**Definition 3.2** (Inversion Robustness). Given an image $x_0$, a text condition $c$, and a stochasticity ratio $\eta$, we define the inversion robustness as follows:

$$\text{IR}(x_0, c, \eta) = d(x_0, \mathbf{Gen}(\mathbf{Inv}(x_0, c), c, \eta)) \quad (5)$$

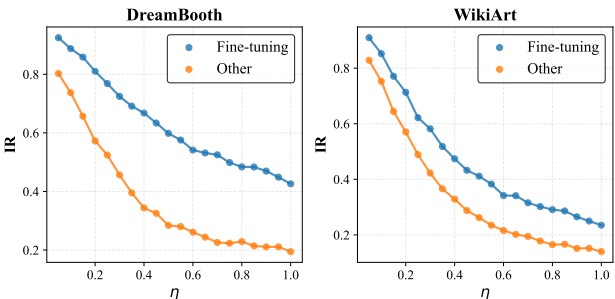

*Figure 3.* Variation of inversion robustness (IR) with different stochasticity ratios $\eta$ on the DreamBooth and WikiArt datasets. At any stochasticity ratio $\eta$, images from the fine-tuning set exhibit significantly higher similarity (SSCD) to their reconstructed counterparts than other images.

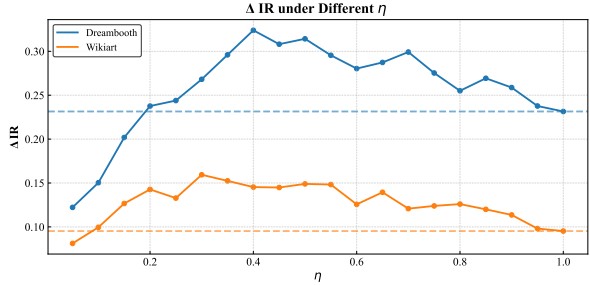

*Figure 4.* The gap in IR between fine-tuning and other samples ($\Delta$IR) varies with the stochasticity ratio $\eta$ on the DreamBooth and WikiArt datasets. When $\eta$ is in the intermediate range, $\Delta$IR is larger than that at $\eta$ close to 0 (a standard inversion–reconstruction process) or $\eta$ close to 1 (a standard generation process).

where **IR** denotes inversion robustness and $d$ denotes a similarity metric (which will be specified later). Based on this definition, we further analyze the inversion robustness for images from the fine-tuning set and those outside it.

Motivated by the above analysis and the visual results in Fig. 2, we reveal a significant difference in inversion robustness between fine-tuning images and other images. Formally, we propose the following assumption.

**Assumption 3.3** (Discrepancy in Inversion Robustness). Given an image $x_{\mathrm{mem}}$ from the fine-tuning set, an image $x_{\mathrm{out}}$ that is not part of the fine-tuning set, and a fine-tuned text condition $c_{\mathrm{ft}}$, We hypothesize that the following holds for any stochasticity ratio $\eta \in [0, 1]$:

$$\underbrace{\mathbb{E}_{x_{\mathrm{mem}}}\left[\mathbf{IR}(x_{\mathrm{mem}}, c_{\mathrm{ft}}, \eta)\right]}_{\textbf{Fine-tuning}} > \underbrace{\mathbb{E}_{x_{\mathrm{out}}}\left[\mathbf{IR}(x_{\mathrm{out}}, c_{\mathrm{ft}}, \eta)\right]}_{\textbf{Other}} \quad (6)$$

Empirically, we validate this assumption by using Self-Supervised Descriptor (SSCD) score (Pizzi et al., 2022) as the metric $d$, a metric designed to detect and quantify copying in diffusion models (Wen et al., 2024; Wu et al., 2025). We compute the two metrics defined in Eq. (6) on Stable Diffusion v1.5 (RunwayML, 2022), fine-tuned on the DreamBooth dataset (Ruiz et al., 2023) and WikiArt subset (Saleh & Elgammal, 2015). As illustrated in Fig. 3, the results show that, across both datasets, images from the fine-tuning set exhibit significantly higher inversion robustness than other images, thereby validating our Assumption 3.3. We also provide detailed experimental settings, additional similarity metrics $d$, and a theoretical analysis of the model's single-step prediction error to further validate Assumption 3.3 in App. B.

### 3.4. Verification Pipeline

In this section, we introduce the verification pipeline of RECOVER, an effective non-intrusive detection method against unauthorized data usage that does not rely on access

to the pre-fine-tuned model.

---
**Algorithm 1** Overview of RECOVER
---
**Input:** Image $x$, model parameter $\theta$, stochasticity ratio list $L = [\eta_1, \eta_2, \ldots, \eta_n]$ of length $n$, text condition $c$, number of reconstructed images $m = k \times n$.
**Output:** Reconstructed images $(x^1, x^2, \ldots, x^m)$

1  $z_T \leftarrow \mathbf{Inv}(x, c)$ in Eq.(1) ;
   **for** $i \leftarrow 1$ **to** $n$ **do**
2       $\eta_i \leftarrow L[i]$ ;
     **for** $j \leftarrow 1$ **to** $k$ **do**
3            $x^{(i-1)\times k+j} \leftarrow \mathbf{Gen}(z_T, c, \eta_i)$ in Eq.(2)
4  **return** $(x^1, x^2, \ldots, x^m)$

---

**Overview**. Algorithm 1 illustrates the overall framework of our method. First, the given image $x$ is inverted into its latent code $z_T$. Next, the image is reconstructed under a series of predefined stochasticity ratios $L$. Finally, potential infringement is determined based on the similarity between the reconstructed images and the given image. According to Assumption 3.3, a high similarity indicates that the given image $x$ was likely used in the fine-tuning process, while a substantial discrepancy is expected if it was not.

**Choice of the stochasticity ratio $\eta$.** Although fine-tuning images consistently exhibit stronger inversion robustness than other images across all $\eta$ values, the strong generalization ability of T2I diffusion models reduces this gap under extreme conditions. When $\eta$ is too low (i.e., a standard inversion–reconstruction process), real images can still be well reconstructed. When $\eta$ is too high (i.e., a standard generation process), the generated images become highly diverse. Both cases diminish the difference in inversion robustness. Therefore, we selectively sample intermediate $\eta$ values to ensure a more pronounced IR distinction. Based on the experimental results in Fig. 3, we further compute the gap in IR between fine-tuning and other samples on both the DreamBooth and WikiArt datasets, defined

as $\Delta \mathbf{IR} = \mathbb{E}_{x_{\mathrm{mem}}}\left[\mathbf{IR}(x_{\mathrm{mem}}, c_{\mathrm{ft}}, \eta)\right] - \mathbb{E}_{x_{\mathrm{out}}}\left[\mathbf{IR}(x_{\mathrm{out}}, c_{\mathrm{ft}}, \eta)\right]$. The results are presented in Fig. 4. As illustrated, $\Delta\mathbf{IR}$ becomes larger when $\eta$ is not at either of the two extreme cases. Therefore, we select $\eta \in [0.3, 0.7]$, where $\Delta IR$ achieves a relatively large distinction for both subject-level fine-tuning and style-level fine-tuning. We also provide a theoretical analysis in the App. C to further justify the choice of $\eta$.

## 4. Experiment

### 4.1. Experiment Setup

**Dataset.** We follow the experimental setup used in previous non-intrusive detection methods (Wu et al., 2024; 2025). For style-driven generation, we construct a WikiArt subset by randomly selecting 20 artworks from each of 20 artists in the WikiArt dataset (Saleh & Elgammal, 2015). For subject-driven generation, we employ the DreamBooth dataset (Ruiz et al., 2023), which contains 30 subjects, each with more than 4 images. Half of the images for each subject are used for fine-tuning, while the remaining half are excluded from training. Consequently, 10 images are used to fine-tune a single artistic style, and 2-3 images are used to fine-tune a single object, consistent with the recommended number of images in fine-tuning methods (Ruiz et al., 2023; Hu et al., 2022).

**Pre-trained Models.** Six mainstream text-to-image diffusion models (*i.e.*, Stable Diffusion v1.4 (Rombach et al., 2022), Stable Diffusion v1.5 (RunwayML, 2022), Stable Diffusion v2.0 (AI, 2022a), Stable Diffusion v2.1 (AI, 2022b), AltDiffusion (Ye et al., 2024) and Stable Diffusion XL (Podell et al., 2023)) are used in the experiments with Stable Diffusion v1.5 set as the default model. It should be emphasized that Stable Diffusion XL and the Stable Diffusion variants (v1.x and v2.x) are built upon different model architectures. To further evaluate the performance of our method on modern architectures, we provide additional evaluations of two state-of-the-art models (Stable Diffusion v3 (Esser et al., 2024) and FLUX (Labs, 2024) ) in the App. E.4. Notably, the noise prediction networks employed by these models are based on Diffusion Transformers, and they utilize flow matching for both training and sampling. This represents a significant departure from traditional U-Net architectures and Denoising Diffusion Probabilistic Models. Testing on these contemporary models further demonstrates the generalizability and practical utility of our approach.

**Fine-tuning Details.** We employ three popular fine-tuning methods, including LoRA (Hu et al., 2022), DreamBooth (with prior loss) (Ruiz et al., 2023), and DreamBooth (without prior loss) (Ruiz et al., 2023) with LoRA serving as our default configuration. We provide additional fine-tuning details in the App. D.1.

**Evaluation Metrics.** For each original image and its re-

constructed counterpart, we compute their visual similarity. Following prior studies (Wen et al., 2024; Wu et al., 2025), we adopt SSCD similarity (Pizzi et al., 2022) as our default similarity metric. Following prior study (Wu et al., 2024), after obtaining the similarity scores, we determine an optimal threshold to distinguish between fine-tuning and other samples. We then compute the Accuracy (ACC) and the Area Under the ROC Curve (AUC) to evaluate the discriminative effectiveness between the two categories, thereby assessing the performance of the signal in copyright authentication. Notably, we use a fixed threshold across different objects and styles. This choice better reflects real-world scenarios, where determining the optimal threshold for each object or style in advance is impractical, especially when the pre-fine-tuned model is unavailable. In practical use, we employ a fixed threshold to distinguish fine-tuning from non-fine-tuning images, with stricter criteria enforced by using a higher threshold. For a more comprehensive evaluation, we also report results obtained by setting optimal thresholds separately for each object or style in the App. E.

**Baselines.** Only two studies (*i.e.*, CGI-DM (Wu et al., 2024) and FineXtract (Wu et al., 2025)) have focused on non-intrusive detection against unauthorized data usage in few-shot generation. Specifically, CGI-DM (Wu et al., 2024) leverages the gradient differences between the pre-fine-tuned and fine-tuned models to reconstruct corrupted images, using the similarity between the reconstructed and original images as an infringement signal. FineXtract (Wu et al., 2025) exploits the differences in predicted noise between the pre-fine-tuned and fine-tuned models to extract images from the fine-tuning set, using the similarity between the extracted and original images as an infringement signal. Since both baselines require access to the pre-fine-tuned model, we follow CGI-DM (Wu et al., 2024) and adopt two image generation pipelines (*i.e.*, Text2img and Img2img) as potential baselines without pre-fine-tuned model. We provide the details of these baselines in the App. D.2.

**Implementation Details.** We set the stochasticity ratio list as $L = [0.35, 0.4, 0.5, 0.6, 0.65]$ with a length of 5. For each stochasticity ratio, we reconstruct one image by default and take the average similarity across all reconstructions as the final score. Consequently, a total of five reconstructed images are generated for each given image. This process requires only six sampling steps in total (including inversion), making our method highly efficient with minimal computational and resource overhead. A detailed analysis of runtime and computational resources is provided in the App. F.

### 4.2. Comparison with Existing Methods

In this section, we compare the effectiveness of our method with existing approaches. As shown in Tab. 1, we conduct subject-driven generation experiments on the DreamBooth

*Table 1.* Comparison of RECOVER and baseline methods on subject-driven generation across multiple base models using LoRA. The experimental results demonstrate that RECOVER exhibits superior performance under most scenarios. Best performance is **bolded**.

| | Pre-fine-tuned Model Free | Stable Diffusion v1.5 | | Stable Diffusion v1.4 | | Stable Diffusion v2.1 | | Stable Diffusion v2.0 | |
|---|---|---|---|---|---|---|---|---|---|
| **Subject-Driven Generation: DreamBooth Dataset** | | ACC | AUC | ACC | AUC | ACC | AUC | ACC | AUC |
| CGI-DM (Wu et al., 2024) | ✗ | 86.71% | 93.81% | 86.08% | 92.19% | **87.34%** | **93.37%** | 86.71% | 90.50% |
| FineXtract (Wu et al., 2025) | ✗ | 73.42% | 78.89% | 69.62% | 73.07% | 71.52% | 78.46% | 68.35% | 74.16% |
| Text2img | ✔ | 67.62% | 73.78% | 65.82% | 69.35% | 66.46% | 70.84% | 66.45% | 70.56% |
| Img2img | ✔ | 65.82% | 68.30% | 66.46% | 67.51% | 60.76% | 63.14% | 64.56% | 64.02% |
| RECOVER | ✔ | **89.87%** | **94.23%** | **87.97%** | **92.84%** | **87.34%** | 92.14% | **87.97%** | **90.90%** |

*Table 2.* Comparison of RECOVER and baseline methods on style-driven generation across multiple base models using LoRA. The experimental results demonstrate that RECOVER exhibits superior performance under most scenarios. Best performance is **bolded**.

| | Pre-fine-tuned Model Free | Stable Diffusion v1.5 | | Stable Diffusion v1.4 | | Stable Diffusion v2.1 | | Stable Diffusion v2.0 | |
|---|---|---|---|---|---|---|---|---|---|
| **Style-Driven Generation: WikiArt Dataset** | | ACC | AUC | ACC | AUC | ACC | AUC | ACC | AUC |
| CGI-DM (Wu et al., 2024) | ✗ | 78.50% | 83.79% | 77.50% | 83.59% | **83.00%** | 87.00% | 80.00% | 82.27% |
| FineXtract (Wu et al., 2025) | ✗ | 68.00% | 69.21% | 69.50% | 70.82% | 64.00% | 67.33% | 70.50% | 69.58% |
| Text2img | ✔ | 65.00% | 63.70% | 63.00% | 64.25% | 63.00% | 64.83% | 69.00% | 68.10% |
| Img2img | ✔ | 75.00% | 76.40% | 74.50% | 79.75% | 74.50% | 80.17% | 74.00% | 78.91% |
| RECOVER | ✔ | **84.50%** | **90.58%** | **78.00%** | **84.54%** | **83.00%** | **87.03%** | **81.00%** | **86.60%** |

dataset using LoRA fine-tuning across various base models. The results demonstrate that our method outperforms existing baselines across most base models, achieving high performance with an ACC of 87% and an AUC of 90%. Similarly, as presented in Tab. 2, we conduct style-driven generation experiments on the WikiArt dataset, where our method outperforms the baselines across most base models, achieving consistently high ACC and AUC.

Notably, in the more realistic setting where the pre-fine-tuned model is unavailable, our method is the only one that remains highly effective, significantly outperforming the two baselines without access to the pre-fine-tuned model. Even under the unrealistic assumption that the pre-fine-tuned model is available, our approach still achieves comparable performance to the sota methods with pre-fine-tuned model, further demonstrating its effectiveness. Experiments on other base models are presented in the App. E.2.

### 4.3. Generalization

In this section, we assess the generalization of our approach by examining its performance under different fine-tuning techniques and varying numbers of fine-tuning steps. We also include the Text-to-Image pipeline as a baseline, representing a strong approach that does not require access to the pre-fine-tuned model.

**Fine-tuning Methods.** Potential infringers may employ a variety of fine-tuning strategies to train T2I diffusion models. Therefore, in addition to LoRA (Hu et al., 2022), we

*Table 3.* Generalization of RECOVER across different fine-tuning strategies compared with the Text-to-Image baseline. Experiments were conducted on Stable Diffusion v1.5 using 15 randomly selected classes from the DreamBooth dataset.

| Method | Fine-tuning Method | ACC | AUC |
|---|---|---|---|
| Text2img | LoRA | 68.42% | 71.22% |
| | DreamBooth (with prior loss) | 72.37% | 73.62% |
| | DreamBooth (without prior loss) | 75.00% | 77.49% |
| RECOVER | LoRA | 97.37% | 99.43% |
| | DreamBooth (with prior loss) | 98.68% | 99.43% |
| | DreamBooth (without prior loss) | 100% | 100% |

evaluate three other fine-tuning methods on 15 classes from the DreamBooth dataset: DreamBooth (with prior loss) (Ruiz et al., 2023), and DreamBooth (without prior loss) (Ruiz et al., 2023). As presented in Tab. 3, across all fine-tuning methods, our approach consistently achieves high ACC (97%) and AUC (99%) and significantly outperform the Text2img, demonstrating that our method remains effective across a wide range of personalization techniques.

**Fine-tuning Steps.** In a real-world scenario, an infringer may fine-tune a T2I diffusion model using different fine-tuning steps and a smaller number of steps may make it more difficult to reconstruct images from the fine-tuning set. We conduct experiments on 15 classes from the DreamBooth dataset using LoRA (Hu et al., 2022) with varying fine-tuning steps. As illustrated in Fig. 5, RECOVER achieves high ACC (85%) and AUC (90%) even with only 50 fine-tuning steps per image, whereas the Text2img fails to reach

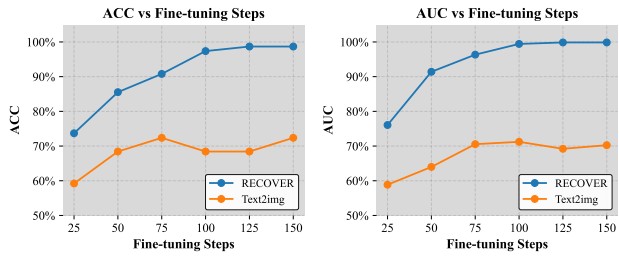

*Figure 5.* Generalization of RECOVER compared with the Text2img using LoRA fine-tuning across different numbers of fine-tuning steps. Other experimental settings follow Tab. 3.

*Table 4.* Influence of stochasticity ratio list and the number of reconstructed images per stochasticity ratio $k$ on the performance of RECOVER. All other settings are kept the same as in Tab. 3.

| Number | $L_1$ | | $L_2$ | | $L_3$ | |
|---|---|---|---|---|---|---|
| | ACC | AUC | ACC | AUC | ACC | AUC |
| 1 | 97.37% | 99.43% | 96.05% | 99.00% | 98.68% | 99.50% |
| 2 | 98.68% | 99.64% | 97.37% | 99.21% | 98.68% | 99.64% |
| 3 | 98.68% | 99.64% | 97.37% | 99.35% | 98.68% | 99.71% |
| 4 | 98.68% | 99.64% | 96.05% | 99.28% | 98.68% | 99.78% |

comparable performance. This also suggests that the target model has not fully captured the target concepts when the number of fine-tuning steps is fewer than 50, which likely contradicts an infringer's intention to effectively reproduce these concepts. Therefore, it is unlikely that infringers would fine-tune for fewer than 50 steps. Visualizations of this experiment are provided in the App. H.2.

**Cross-model Transferability of Thresholds.** In real-world scenarios, we may not always be able to obtain the thresholds of all models in advance. Therefore, in this experiment, we evaluate the cross-model transferability of thresholds, meaning the capability of a threshold computed on one model to perform discrimination on another model. We include a more detailed experimental setup, with the experimental results provided in App. E.3.

### 4.4. Ablation Study

**Different stochasticity ratio list $L$.** We also investigate the influence of different stochasticity ratio lists $L$ and reconstruction numbers per image $k$ on our method. Specifically, we consider three stochasticity ratio lists: $L_1 = [0.35, 0.4, 0.5, 0.6, 0.65]$, $L_2 = [0.4, 0.5, 0.6]$, and $L_3 = [0.3, 0.35, 0.4, 0.5, 0.6, 0.65, 0.7]$, along with four reconstruction counts per image $k = 1, 2, 3, 4$. As illustrated in Tab. 4, the results show that neither the stochasticity ratio list nor the number of reconstructed images has a significant impact on performance, and our method consistently achieves high ACC and AUC across all settings.

**Comparisons Under Identical Similarity Metrics.** To ensure a strictly faithful reproduction of the baselines, our

*Table 5.* Ablation study of the SSCD metric on Stable Diffusion v1.5, with other experimental settings identical to those in Tab. 1. Experimental results indicate that the choice of metric only marginally improves effectiveness, demonstrating that the performance gains stem from the proposed method itself rather than the specific metric design.

| Method | ACC | AUC |
|---|---|---|
| Text2Img | 71.51% | 75.68% |
| Img2Img | 69.62% | 72.37% |
| FineXtract (Wu et al., 2025) | 73.42% | 78.89% |
| CGI-DM (Wu et al., 2024) | 88.61% | 94.06% |
| **RECOVER** | **89.87%** | **94.23%** |

main experiments employed the exact similarity metrics specified in their respective papers and source code. In this specific experiment, to further evaluate the superiority of our inversion robustness, we compared our approach against the baselines using an identical similarity metric (SSCD). This evaluation was conducted on Stable Diffusion v1.5 utilizing 30 classes from the DreamBooth dataset. As illustrated in Tab. 5, the baseline methods exhibit only marginal improvements under this metric and remain significantly inferior to our approach. This further demonstrates that the success of RECOVER stems from utilizing a stronger signal to distinguish whether an image has been used for fine-tuning, rather than selecting a more advantageous similarity metric.

### 4.5. Robustness

In this section, we assume that an infringer is aware that we may employ copyright authentication methods to assert ownership. To circumvent such authentication, the infringer may adopt two strategies: (i) pre-processing the dataset, and (ii) limited data ratios.

**Dataset Pre-processing.** Pre-processing the fine-tuning dataset makes it more challenging to reconstruct the original fine-tuning images using the given model. Therefore, we conduct a dataset pre-processing experiment on 15 categories from the DreamBooth dataset, applying four types of pre-processing operations: Noise, Blur, JPEG compression, and Sharpening. As shown in Tab. 6, even under strong perturbations (e.g., JPEG 35), RECOVER consistently achieves high ACC and AUC across various pre-processing methods, demonstrating its robustness against dataset pre-processing. Such consistency across diverse pre-processing types suggests that our method remains a reliable auditing tool for real-world copyright protection.

**Limited Data Ratios.** Potential infringers may fine-tune diffusion models on mixed datasets that combine the target fine-tuning data with other public or unrelated images, thereby reducing the proportion of target data in training. To simulate this scenario, we conduct experiments on 15 subjects from the DreamBooth dataset, following the same

*Table 6.* Robustness of RECOVER against image pre-processing. All other experimental settings are kept the same as in Tab. 3.

| Pre-processing Method | Setting | ACC | AUC |
|---|---|---|---|
| Noise | $\sigma = 0.3$ | 94.74% | 99.00% |
| Blur | $k = 7$ | 96.05% | 98.85% |
| JPEG | $q = 35$ | 94.74% | 98.06% |
| Sharpen | $\alpha = 5$ | 96.05% | 98.21% |

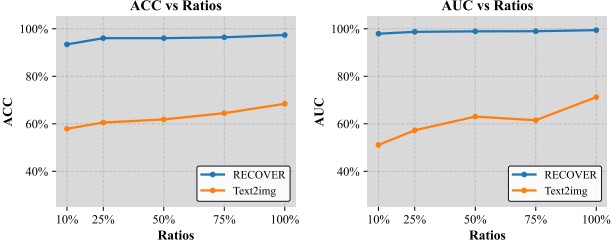

*Figure 6.* Robustness of RECOVER compared with the Text2img using LoRA fine-tuning against limited data ratios. All other experimental settings follow Tab. 3.

setup as in Tab. 3. For each subject, we set the proportion of its images in the fine-tuning dataset to 10%, 25%, 50%, 75%, and 100%, respectively, and fill the remaining portion with images randomly sampled from other subjects in the DreamBooth dataset. For example, if a subject has 3 images and the proportion is set to 10%, we include those 3 images and randomly sample an additional $3/10\% - 3 = 27$ images from other subjects to complete the fine-tuning dataset. As shown in Fig. 6, RECOVER maintains high ACC and AUC even when the proportion of target images is as low as 10%, whereas the performance of the Text2Img degrades significantly under the same condition. This indicates that when the proportion of target data falls below 10%, the model fails to effectively learn the target concepts, contradicting an infringer's original intent to capture these concepts through fine-tuning. Visualization results of this experiment are provided in the App. H.3.

**Potential Countermeasures.** To further evaluate the robustness of our method against potential countermeasures, we investigate four types of adaptive attacks: (1) Training-time Augmentations, which apply random masks during fine-tuning to disrupt the continuity of the learned denoising trajectories; (2) Trajectory Sensitivity Regularization (TSR), which introduces an additional loss term to maximize the deviation between the original and perturbed noise predictions; (3) LoRA Rank Scaling, which varies the capacity of fine-tuning parameters; and (4) Trigger Prompt Distortion, which tests the method using LLM-generated variant prompts. A comprehensive discussion regarding these adaptive attacks, including detailed experimental setups and full results, is provided in the App. E.5.

## 5. Conclusion

In this paper, we present, for the first time, a reliable non-intrusive detection method that operates without access to the pre-fine-tuned model to combat unauthorized data usage. By analyzing the training objectives of diffusion models, we reveal that fine-tuning images exhibit more stable reconstruction under noise perturbations, a property we term *inversion robustness*. Building upon this insight, we propose RECOVER, a method that leverages inversion robustness to detect unauthorized data usage without requiring intrusive modifications or the pre-fine-tuned model. Extensive experiments demonstrate the effectiveness, generalizability, and robustness of our approach. We hope future work will further explore detecting unauthorized data usage under such constrained yet realistic scenarios.

## Impact Statement

The rapid advancement of generative AI has raised significant concerns regarding unauthorized data usage and copyright infringement. While our proposed method, RECOVER, empowers content creators to protect their intellectual property by providing reliable technical auditing evidence, we recognize that such technologies are a double-edged sword. To mitigate the risk of over-enforcement that could inadvertently stifle fair use and hinder AI innovation, RECOVER is strictly designed as a technical auditing aid rather than an automated weapon for punitive actions. Ultimately, our goal is to foster a responsible AI ecosystem by striking a crucial balance between safeguarding creator rights and preserving the collaborative spirit essential for technological progress.

## Acknowledgement

This research was supported in part by the National Natural Science Foundation of China (NSFC) under Grants No. 62576255, the Natural Science Foundation of Hubei Province under No. 2025AFB455.

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

# A. Detailed Discussion on Existing Methods

## A.1. Existing Non-intrusive Detection Methods

In this section, we discuss a realistic scenario where the pre-fine-tuned model is maliciously claimed by the infringer, under which the effectiveness of existing non-intrusive detection methods significantly deteriorates.

Such a scenario is practical in legal disputes, where the infringer can freely submit any model as the alleged pre-fine-tuned version. Existing methods such as CGI-DM (Wu et al., 2024) and FineXtract (Wu et al., 2025) require both the pre-fine-tuned model $\theta'$ and the suspect fine-tuned model $\theta$ to be provided by the claimant. However, without any prior knowledge about the authentic pre-fine-tuned model, it is impossible to verify whether the submitted $\theta'$ is genuine.

Since these approaches rely on the discrepancy between $\theta'$ and $\theta$. Their effectiveness sharply decreases when the model discrepancy becomes smaller.

To empirically validate this observation, we conduct experiments on the Dreambooth dataset using LoRA fine-tuning. Specifically, we simulate malicious claims by using models fine-tuned with different numbers of steps as the alleged pre-fine-tuned models. A larger number of steps indicates a smaller difference between the claimed and suspect models. As shown in Fig. 7, the detection performance of CGI-DM (Wu et al., 2024) and FineXtract (Wu et al., 2025) consistently degrades as the fine-tuning steps increase, and when the maliciously claimed model is fine-tuned for 75 steps, their performance even falls below that of direct text-to-image generation.

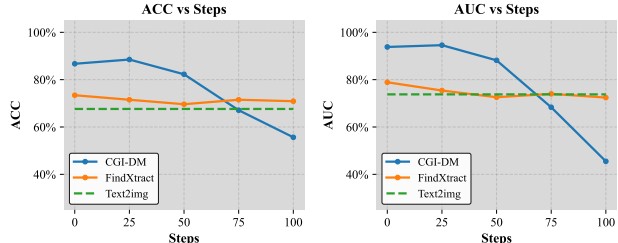

*Figure 7.* Results of existing non-intrusive detection methods when the pre-fine-tuned model is maliciously claimed.

## A.2. Membership Inference

Membership inference (Shokri et al., 2017; Carlini et al., 2022; Duan et al., 2023; Dubiński et al., 2024; Pang et al., 2024; Kong et al., 2024) was originally proposed as a privacy leakage attack, aiming to determine whether a specific data sample was included in the training set of a model. In the context of copyright protection, such techniques may serve as potential non-intrusive approaches to detect unauthorized data usage. However, applying membership inference to identify unauthorized data usage faces several limitations.

In a standard membership inference pipeline, two main stages are involved: the attack model acquisition stage and the inference stage (the attack model is applied). During the acquisition stage, the attacker first trains one or more shadow models using the pre-fine-tuned model, and then trains a attack model based on the shadow model's outputs to distinguish between fine-tuning and non-fine-tuning samples. However, this process requires access to the pre-fine-tuned model, suffering from the same limitation as existing non-intrusive detection methods discussed in Sec. 2.2.

On the other hand, existing non-intrusive methods (Wu et al., 2024; 2025) and RECOVER can provide **visual evidence**, which is more intuitive for human interpretation and serves as a stronger legal proof compared to binary classification results (Wu et al., 2024). Compared with binary classification outputs, **visual similarity** is much easier for humans to interpret, as vision is one of the most fundamental human perceptual abilities and people possess strong priors for judging visual similarity. Consequently, evidence that is broadly understandable is more likely to be widely accepted.

# B. Details for Assumption 3.3

## B.1. Details of Empirical Evaluation

In this section, we provide the details of our empirical evaluation conducted to validate Assumption 3.3.

For subject-driven generation, we use the DreamBooth dataset (Ruiz et al., 2023) , which contains 30 subjects with 4–6 images per subject. For each subject, half of the images are used as the fine-tuning set to train the model, while the remaining images serve as the non-fine-tuning set. Similarly, for style-driven generation, we use a subset of the WikiArt dataset (Saleh & Elgammal, 2015) containing 20 artists, with 20 images per artist. For each artist, 10 images are used as the fine-tuning set,

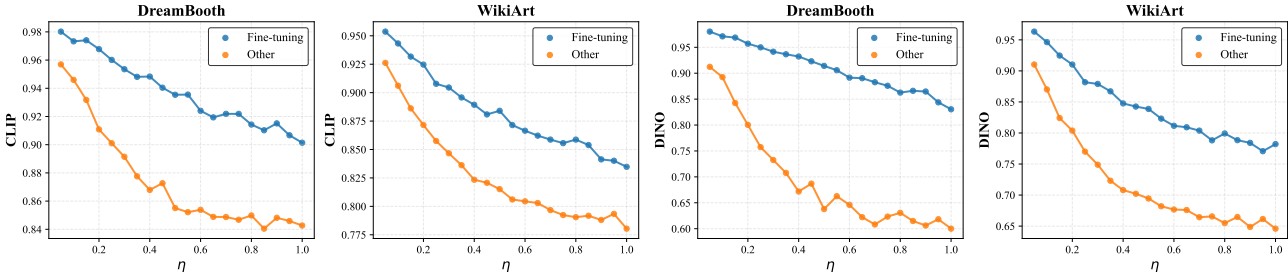

*Figure 8.* Variation of inversion robustness (IR) with different stochasticity ratios $\eta$ on the DreamBooth and WikiArt datasets. At any stochasticity ratio $\eta$, images from the fine-tuning set exhibit significantly higher similarity (CLIP and DINO) to their reconstructed counterparts than other images.

while the remaining 10 images serve as the non-fine-tuning set.

After preparing the datasets, we fine-tune 30 subject models and 10 style models using LoRA (Hu et al., 2022). For each model, we measure the inversion robustness of its fine-tuning and non-fine-tuning images across 20 stochasticity ratios. Finally, after obtaining all scores for the fine-tuning and non-fine-tuning sets, we compute the average score across the 20 stochasticity ratios for each set.

## B.2. Other Distance Metrics

To validate the generality of Assumption 3.2, we employ a broader set of distance metrics $d$ as defined in Eq. (5), including the s CLIP similarity (Radford et al., 2021), DINO similarity (Oquab et al., 2024).

As shown in Fig. 8, Assumption 3.3 consistently holds across all evaluated metrics for $d$, thereby confirming the applicability of the discrepancy in inversion robustness.

## B.3. Theoretical Analysis of Single-Step Error

In this section, we provide a formal analysis of the single-step prediction error in inversion robustness. By analyzing the relationship between the forward noise, the inversion trajectory, and additional sampling noise, we demonstrate that the reconstruction error for training set samples is inherently smaller than that for out-of-training samples, thereby providing a theoretical foundation for Assumption 3.3.

**Proposition B.1.** *Let $x_t(\sigma_t)$ be the intermediate latent state obtained by combining DDIM inversion noise and an additional random perturbation scaled by $\sigma_t > 0$. Due to the generalization gap inherent in the diffusion model's optimization, the expected squared single-step reconstruction error is strictly smaller for samples from the training distribution $q_{\mathrm{mem}}$ than for those from the out-of-training distribution $q_{\mathrm{out}}$:*

$$\mathbb{E}_{x \sim q_{\mathrm{mem}}} \|\hat{x}_0 - x_0\|^2 < \mathbb{E}_{x \sim q_{\mathrm{out}}} \|\hat{x}_0 - x_0\|^2. \tag{7}$$

*Proof.* We first formulate the forward diffusion process with explicit dependence on the perturbation scale $\sigma_t$. Let the perturbed latent state be defined as:

$$x_t(\sigma_t) = \sqrt{\alpha_t}\, x_0 + \sqrt{1 - \alpha_t}\, \epsilon_{\mathrm{true}}, \tag{8}$$

$$\sqrt{1 - \alpha_t}\, \epsilon_{\mathrm{true}} = \sqrt{1 - \alpha_t - \sigma_t^2}\, \epsilon_{\mathrm{inv}} + \sigma_t\, \epsilon_t, \tag{9}$$

where $\epsilon_{\mathrm{inv}}$ is the noise trajectory obtained from DDIM inversion. We assume $\epsilon_{\mathrm{inv}} \sim \mathcal{N}(0, I)$ and is independent of the additional sampling noise $\epsilon_t \sim \mathcal{N}(0, I)$. This assumption ensures that the combined true forward noise $\epsilon_{\mathrm{true}}$ remains standard normal, thereby satisfying the marginal distributions of the forward process.

We define the model's noise prediction error at the perturbed state $x_t(\sigma_t)$ as:

$$\delta_{\theta,t}(\sigma_t) := \epsilon_\theta(x_t(\sigma_t), t) - \epsilon_{\mathrm{true}}. \tag{10}$$

Using the DDIM formulation, the single-step prediction of the clean image is given by:

$$\hat{x}_0 = \frac{1}{\sqrt{\alpha_t}}\big(x_t(\sigma_t) - \sqrt{1-\alpha_t}\,\epsilon_\theta(x_t(\sigma_t),t)\big). \tag{11}$$

Substituting $x_t(\sigma_t)$ into the above equation yields the single-step reconstruction error:

$$\hat{x}_0 - x_0 = \frac{\sqrt{1-\alpha_t}}{\sqrt{\alpha_t}}\left(\epsilon_{\text{true}} - \epsilon_\theta(x_t(\sigma_t),t)\right) = -\sqrt{\frac{1-\alpha_t}{\alpha_t}}\,\delta_{\theta,t}(\sigma_t). \tag{12}$$

To rigorously understand the dependence of $\delta_{\theta,t}$ on $\sigma_t$, let $x_t^{\text{inv}}$ denote the exact inversion state when $\sigma_t = 0$. For a small $\sigma_t > 0$, applying a first-order Taylor expansion to the neural network $\epsilon_\theta$ around $x_t^{\text{inv}}$ yields:

$$\epsilon_\theta(x_t(\sigma_t),t) \approx \epsilon_\theta(x_t^{\text{inv}},t) + \nabla_x\epsilon_\theta(x_t^{\text{inv}},t) \cdot (x_t(\sigma_t) - x_t^{\text{inv}}). \tag{13}$$

As $\sigma_t$ increases, the state $x_t(\sigma_t)$ deviates from the deterministic inversion path $x_t^{\text{inv}}$ by random noise. To accurately estimate the noise on this perturbed manifold, the model relies on the gradient $\nabla_x\epsilon_\theta$, which is governed by the data distribution learned during training.

Taking the expectation of the squared reconstruction error, we have:

$$\mathbb{E}_{x\sim q}\|\hat{x}_0 - x_0\|^2 = \frac{1-\alpha_t}{\alpha_t}\,\mathbb{E}_{x\sim q}\|\delta_{\theta,t}(\sigma_t)\|^2, \tag{14}$$

where $q$ represents either the training set $q_{\text{mem}}$ or the out-of-training set $q_{\text{out}}$.

Because the diffusion model is trained via Empirical Risk Minimization (ERM) on $q_{\text{mem}}$, it inherently memorizes the training data manifold, resulting in a generalization gap. Consequently, for perturbed inputs ($\sigma_t > 0$), the model's prediction error on unseen data is strictly bounded above the error on the training data. This generalization gap dictates that:

$$\mathbb{E}_{x\sim q_{\text{mem}}}\|\delta_{\theta,t}(\sigma_t)\|^2 < \mathbb{E}_{x\sim q_{\text{out}}}\|\delta_{\theta,t}(\sigma_t)\|^2. \tag{15}$$

Substituting this inequality back into the expected reconstruction error equation, we arrive at the final conclusion:

$$\mathbb{E}_{x\sim q_{\text{mem}}}\|\hat{x}_0 - x_0\|^2 < \mathbb{E}_{x\sim q_{\text{out}}}\|\hat{x}_0 - x_0\|^2. \tag{16}$$

This concludes the proof.

$\square$

**Remark on Theoretical Scope.** While the above proof provides a formal foundation for the single-step prediction error, we acknowledge that extending this analysis to derive strict mathematical bounds for the full, multi-step generation trajectory remains notoriously difficult. Modern diffusion architectures, such as U-Net and DiT, are highly non-linear, making end-to-end theoretical formalization an open challenge for the broader community. Therefore, we emphasize that the primary contribution of RECOVER is fundamentally empirical and application-driven.

Nevertheless, rather than leaving the effectiveness of RECOVER as a black-box phenomenon, this single-step analysis offers crucial mathematical intuition. It explicitly demystifies why models exhibit systematically smaller denoising errors on training data under random noise, which directly inspires and grounds our empirical design of the intermediate stochasticity ratio. In a rapidly evolving field where theoretical formalization naturally trails behind empirical advancements, we view this analysis as a meaningful first step to bridge theoretical intuition with empirical observations, hoping it serves as a valuable starting point to inspire future, more rigorous theoretical research.

## C. Theoretical Analysis of the Inversion Robustness Gap

In this section, we provide a formal theoretical proof to explain the empirical observation that the inversion robustness gap between fine-tuning samples and non-fine-tuning samples is maximized at an intermediate stochasticity ratio $\eta \in (0,1)$.

Let $x_0$ denote an initial given image. We define the expected reconstruction error under a stochasticity ratio $\eta$ as:

$$E(\eta) = \mathbb{E}\left[\|Gen(Inv(x_0, c), c, \eta) - x_0\|^2\right] \tag{17}$$

where $Inv(\cdot)$ represents the DDIM inversion mapping $x_0 \mapsto z_T$, and $Gen(\cdot, \eta)$ represents the generative process.

We denote the expected reconstruction error for the fine-tuning set as $E_{mem}(\eta) = \mathbb{E}_{x_0 \sim q_{mem}}[\|\hat{x}_0(\eta) - x_0\|^2]$, and for the non-fine-tuning set as $E_{out}(\eta) = \mathbb{E}_{x_0 \sim q_{out}}[\|\hat{x}_0(\eta) - x_0\|^2]$. The inversion robustness gap is defined as the difference in reconstruction errors:

$$\Delta E(\eta) = E_{out}(\eta) - E_{mem}(\eta) \tag{18}$$

Our goal is to prove that $\Delta E(\eta)$ achieves a positive maximum at an intermediate value $\eta^* \in (0, 1)$. To do so, we first establish the boundary conditions at $\eta \to 0$ and $\eta \to 1$.

**Lemma C.1** (Deterministic Inversion Limit). *As the stochasticity ratio approaches zero ($\eta \to 0$), the inversion robustness gap vanishes:* $\lim_{\eta \to 0} \Delta E(\eta) = 0$.

*Proof.* When $\eta = 0$, the DDIM generation process reduces to a deterministic Probability Flow ODE. In the continuous-time limit (assuming negligible discretization error $\tau$), the forward ODE (inversion) and the reverse ODE (generation) are exact inverse mappings. Therefore, regardless of whether $x_0 \sim q_{mem}$ or $x_0 \sim q_{out}$, the trajectory perfectly cycles back to the initial condition, yielding $\lim_{\eta \to 0} E_{mem}(\eta) = \mathcal{O}(\tau)$ and $\lim_{\eta \to 0} E_{out}(\eta) = \mathcal{O}(\tau)$. Consequently, $\lim_{\eta \to 0} \Delta E(\eta) = 0$. $\qquad\square$

**Lemma C.2** (Stochastic Mixing Limit). *As the stochasticity ratio approaches one ($\eta \to 1$), the inversion robustness gap vanishes:* $\lim_{\eta \to 1} \Delta E(\eta) = 0$.

*Proof.* When $\eta = 1$, the generative process operates as a fully stochastic differential equation (SDE), equivalent to standard DDPM sampling. In this regime, the magnitude of the injected noise guarantees that the reverse Markov chain rapidly mixes to its stationary distribution, erasing the specific information embedded in the initial latent code $z_T$. Thus, $\hat{x}_0(1)$ becomes an unconditional sample drawn from the learned distribution $p_\theta(x|c)$, independent of $x_0$.

Let the variance of the generated data distribution be $V_{data}$. The expected error for any input simply converges to the variance between a random sample and the data manifold: $\lim_{\eta \to 1} E_{mem}(\eta) \approx V_{data}$ and $\lim_{\eta \to 1} E_{out}(\eta) \approx V_{data}$. Therefore, $\lim_{\eta \to 1} \Delta E(\eta) = 0$. $\qquad\square$

**Lemma C.3** (Local Error Divergence via Lipschitz Continuity). *For a sufficiently small perturbation $\eta > 0$, the reconstruction error grows strictly faster for out-of-distribution samples than for fine-tuning samples, meaning $\left.\frac{\partial \Delta E(\eta)}{\partial \eta}\right|_{\eta \to 0^+} > 0$.*

*Proof.* Consider a small injected noise bounded by $\eta$. The local trajectory deviation $\delta x_t$ is determined by the score prediction error $\|s_\theta(x_t(\eta), t) - \nabla_x \log p_t(x_t(\eta))\|$. This error is governed by the local Lipschitz constant $L$ of the learned score function $s_\theta$.

Because the diffusion training objective $\mathcal{L}_{DM}$ explicitly minimizes the score matching error for samples in the training manifold $q_{mem}$, the vector field around $q_{mem}$ forms a smooth, robust basin of attraction with a small Lipschitz constant $L_{mem}$. Conversely, the unconstrained region around $q_{out}$ relies solely on generalizable extrapolation, leading to a much steeper score field variation $L_{out} \gg L_{mem}$.

Applying a first-order Taylor expansion to the expected error expectation with respect to the perturbation magnitude $\eta$, we have $\frac{\partial E}{\partial \eta} \propto \mathbb{E}[L]$. Since $L_{out} > L_{mem}$, it follows that:

$$\left.\frac{\partial E_{out}}{\partial \eta}\right|_{\eta \to 0^+} > \left.\frac{\partial E_{mem}}{\partial \eta}\right|_{\eta \to 0^+} > 0 \tag{19}$$

Thus, the derivative of the gap $\frac{\partial \Delta E}{\partial \eta} = \frac{\partial E_{out}}{\partial \eta} - \frac{\partial E_{mem}}{\partial \eta} > 0$ for $\eta \to 0^+$. $\qquad\square$

**Theorem C.4** (Maximum Inversion Robustness Gap). *There exists an intermediate stochasticity ratio $\eta^* \in (0, 1)$ such that the inversion robustness gap $\Delta E(\eta)$ achieves a positive global maximum.*

*Proof.* Let $\Delta E(\eta)$ be a continuous function defined on $[0, 1]$. By Lemma C.1 and Lemma C.2, we have boundary conditions $\Delta E(0) = 0$ and $\Delta E(1) = 0$. By Lemma C.3, $\Delta E(\eta)$ is strictly increasing in the right-neighborhood of $\eta = 0$, meaning there exists some $\eta_0 > 0$ such that $\Delta E(\eta_0) > 0$.

According to the Extreme Value Theorem, a continuous function on a closed interval must attain its maximum. Since the value at $\eta_0 \in (0, 1)$ is strictly greater than the boundaries, the global maximum must occur at an intermediate critical point $\eta^* \in (0, 1)$. This concludes the proof, formally explaining why intermediate values of $\eta$ yield the strongest detection signal. $\square$

## D. Experimental Setup Details

### D.1. Fine-tuning Details

In this section, we provide details of fine-tuning used in the experiments.

**LoRA (Hu et al., 2022).** This method is a parameter-efficient fine-tuning approach that updates only two low-rank matrices instead of the full model weights, and does not require any additional training data. We set the matrix rank to 64, the batch size to 1, the learning rate to $1 \times 10^{-4}$ and fine-tune each image for 100 steps.

**DreamBooth (w. prior loss) (Ruiz et al., 2023).** This method performs full fine-tuning and requires an additional training set to prevent overfitting on the fine-tuning images. In our experiments, we consistently use 30 model-generated images as the additional training set for this method. We set the batch size to 1, the learning rate to $2 \times 10^{-6}$ and fine-tune each image for 100 steps.

**DreamBooth (w/o. prior loss) (Ruiz et al., 2023).** This method also performs full fine-tuning and does not require an additional training set. We set the batch size to 1, the learning rate to $2 \times 10^{-6}$ and fine-tune each image for 100 steps.

For the DreamBooth dataset, we use the default prompt "a photo of sks [object]" for fine-tuning, where [object] denotes a specific category (e.g., "a photo of sks dog" for the dog class). For the WikiArt dataset, we adopt "a painting in sks style" as the default prompt. When applying DreamBooth (with prior loss), we use "a photo of [object]" and "a painting" as the class prompts for the DreamBooth and WikiArt datasets, respectively. During all fine-tuning processes, both the U-Net and the text encoder are fine-tuned.

### D.2. Baselines Details

In this section, we present the implementation details of the baseline methods.

**CGI-DM (Wu et al., 2024).** CGI-DM leverages the differences between the pre-fine-tuned and post-fine-tuned diffusion models to reconstruct the degraded images and determines potential infringement by comparing the similarity between the reconstructed and original images. We reproduce their implementation following the original configuration and use CLIP similarity (Radford et al., 2021) as the image similarity metric, which is consistent with their original setup.

**FineXtract (Wu et al., 2025).** FineXtract leverages the guidance differences between the pre- and post-fine-tuned models to generate images that are more similar to those in the training set. The potential infringement is then determined by evaluating the similarity between the generated and given images. We reproduce their method following the settings described in the original paper and use SSCD similarity (Pizzi et al., 2022) as the similarity metric, consistent with their configuration.

However, these methods all rely on access to the pre-fine-tuned model as a core component. Therefore, following the previous work (Wu et al., 2024), we adopt existing image generation pipelines as competitive baselines without the access to the pre-fine-tuned model. In particular, we compare the following two types of pipelines:

**Text-to-Image Generation.** We use the official Text-to-Image pipeline from Diffusers to generate 10 images for each given image using its corresponding training prompt, with 50 denoising steps during inference.

**Image-to-Image Generation.** We employ the official Image-to-Image pipeline provided by Diffusers to generate 10 images for each given image, using its corresponding training prompt. The inference process is performed with 50 denoising steps and an img2img strength of 0.7.

For each given image, we generate 10 samples for comparison, ensuring that the runtime of the pipelines is comparable to ours for a fair evaluation. Following previous study (Wu et al., 2024), we adopt CLIP similarity as the image similarity metric for both generation processes. The highest similarity score among these generated samples is used as the final

*Table 7.* Comparison of RECOVER and baseline methods on subject-driven generation across multiple base models using LoRA under a discrete threshold. Best performance is **bolded**.

| | Pre-fine-tuned Model Free | Stable Diffusion v1.5 | | Stable Diffusion v1.4 | | Stable Diffusion v2.1 | | Stable Diffusion v2.0 | |
|---|---|---|---|---|---|---|---|---|---|
| **Subject-Driven Generation: DreamBooth Dataset** | | ACC | AUC | ACC | AUC | ACC | AUC | ACC | AUC |
| CGI-DM (Wu et al., 2024) | ✖ | 95.11% | 94.81% | 95.11% | 93.52% | 95.61% | 95.56% | **97.22%** | **97.04%** |
| FineXtract (Wu et al., 2025) | ✖ | 88.17% | 85.83% | 85.83% | 78.70% | 91.06% | 85.56% | 84.50% | 79.26% |
| Text2img | ✔ | 85.78% | 78.98% | 82.89% | 76.48% | 84.00% | 75.93% | 82.33% | 73.70% |
| Img2img | ✔ | 80.17% | 72.59% | 82.44% | 75.19% | 81.72% | 75.19% | 81.06% | 70.56% |
| RECOVER | ✔ | **97.00%** | **97.78%** | **98.22%** | **99.07%** | **98.22%** | **98.70%** | 96.00% | 96.30% |

*Table 8.* Comparison of RECOVER and baseline methods on style-driven generation across multiple base models using LoRA under a discrete threshold. Best performance is **bolded**.

| | Pre-fine-tuned Model Free | Stable Diffusion v1.5 | | Stable Diffusion v1.4 | | Stable Diffusion v2.1 | | Stable Diffusion v2.0 | |
|---|---|---|---|---|---|---|---|---|---|
| **Style-Driven Generation: WikiArt Dataset** | | ACC | AUC | ACC | AUC | ACC | AUC | ACC | AUC |
| CGI-DM (Wu et al., 2024) | ✖ | 88.50% | 89.10% | 89.50% | 91.80% | 90.00% | 91.90% | 86.00% | 86.60% |
| FineXtract (Wu et al., 2025) | ✖ | 78.50% | 77.20% | 78.50% | 77.15% | 75.50% | 73.45% | 77.50% | 75.45% |
| Text2img | ✔ | 75.00% | 69.40% | 74.00% | 71.80% | 73.50% | 70.35% | 75.50% | 72.70% |
| Img2img | ✔ | 84.00% | 84.70% | 86.00% | 86.20% | 83.50% | 86.10% | 84.50% | 86.05% |
| RECOVER | ✔ | **91.50%** | **94.60%** | **90.00%** | **90.50%** | **91.00%** | **93.30%** | **90.00%** | **92.80%** |

similarity score for the given image.

# E. Additional Experimental Results

### E.1. Result under Separate Threshold

In this section, we evaluate the performance of RECOVER under discrete threshold settings. For a more comprehensive assessment, we determine the optimal threshold for each individual subject or style, rather than adopting a fixed threshold. As shown in Tab. 7 and Tab. 8, RECOVER continues to achieve strong performance under this setup, surpassing most baseline methods.

### E.2. Results on Additional Base Models

In this section, we evaluate RECOVER on additional base models (AltDiffusion (Ye et al., 2024) and Stable Diffusion XL (Podell et al., 2023)). As shown in Tab. 9 and Tab. 10, our method also achieves strong performance on these models, significantly outperforming existing baselines. We also report the performance of our method aggregated across all base models (denoted as "all model"), a scenario that more faithfully reflects the use of a single fixed threshold in practice. In this setting, RECOVER still achieves strong performance and significantly outperforms all baselines.

### E.3. Cross-model Transferability of Thresholds

In this section, we detail the cross-model threshold transferability experiment, where we conduct an all-to-all transferability test across SD 1.5, SD 2.1, and SDXL (three distinct architectures) on 30 classes from the DreamBooth dataset. As shown in Fig. 9, the ACC drop of RECOVER is minimal (<3%), proving that the reverse robustness signal is an intrinsic, architecture-agnostic property of fine-tuned

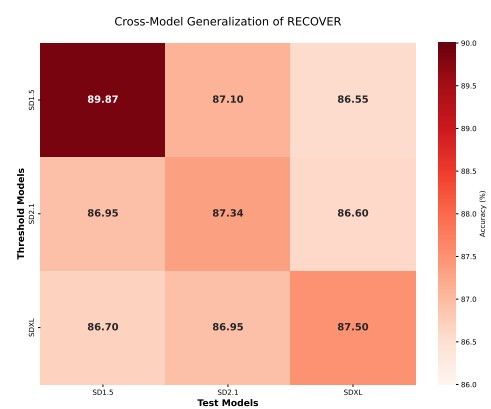

*Figure 9.* Results of the cross-model threshold transferability experiment.

*Table 9.* Comparison of RECOVER and baseline methods on subject-driven generation across additional base models using LoRA. Best performance is **bolded**.

| | Pre-fine-tuned Model Free | Stable Diffusion v1.5 | | Stable Diffusion XL | | AltDiffusion | | All Model | |
|---|---|---|---|---|---|---|---|---|---|
| **Subject-Driven Generation: DreamBooth Dataset** | | | | | | | | | |
| | | ACC | AUC | ACC | AUC | ACC | AUC | ACC | AUC |
| CGI-DM (Wu et al., 2024) | ✘ | 86.71% | 93.81% | 82.53% | 87.81% | 83.54% | 89.93% | 85.06% | 92.14% |
| FineXtract (Wu et al., 2025) | ✘ | 73.42% | 78.89% | 67.62% | 71.98% | 67.97% | 68.50% | 69.43% | 74.75% |
| Text2img | ✔ | 67.62% | 73.78% | 67.09% | 70.98% | 68.99% | 68.38% | 66.08% | 70.73% |
| Img2img | ✔ | 65.82% | 68.30% | 68.35% | 73.02% | 65.61% | 67.25% | 61.34% | 65.69% |
| RECOVER | ✔ | **89.87%** | **94.23%** | **87.97%** | **92.17%** | **87.34%** | **92.86%** | **87.34%** | **92.55%** |

*Table 10.* Comparison of RECOVER and baseline methods on subject-driven generation across additional base models using LoRA. Best performance is **bolded**.

| | Pre-fine-tuned Model Free | Stable Diffusion v1.5 | | Stable Diffusion XL | | AltDiffusion | | All Model | |
|---|---|---|---|---|---|---|---|---|---|
| **Style-Driven Generation: WikiArt Dataset** | | | | | | | | | |
| | | ACC | AUC | ACC | AUC | ACC | AUC | ACC | AUC |
| CGI-DM (Wu et al., 2024) | ✘ | 78.50% | 83.79% | 82.00% | 87.68% | 74.50% | 80.54% | 77.50% | 83.40% |
| FineXtract (Wu et al., 2025) | ✘ | 68.00% | 69.21% | 69.00% | 72.54% | 62.50% | 63.36% | 66.10% | 68.09% |
| Text2img | ✔ | 65.00% | 63.70% | 67.50% | 70.30% | 60.50% | 58.52% | 62.40% | 63.81% |
| Img2img | ✔ | 75.00% | 76.40% | 82.00% | 87.39% | 68.00% | 69.26% | 71.70% | 76.76% |
| RECOVER | ✔ | **84.50%** | **90.58%** | **87.50%** | **92.86%** | **80.50%** | **86.42%** | **80.20%** | **86.97%** |

diffusion models, thereby further demonstrating the practicality of RECOVER.

### E.4. Results on Modern Models

In the main paper, we primarily focus on Stable Diffusion 1.5 to ensure a fair comparison with existing baselines. To demonstrate the generalizability of RECOVER, we previously evaluated its performance on SDXL and AltDiffusion in App. E.2. For a more comprehensive assessment of our method, we further extend our evaluation to frontier modern architectures, including FLUX.1 (Labs, 2024) and Stable Diffusion v3 (Esser et al., 2024).

**Adaptation to Flow Matching Models.** Since RECOVER is built upon the architecture-agnostic property of *Inversion Robustness*, it can be seamlessly applied to Flow Matching frameworks without structural modifications. We simply replace the Stable Diffusion 1.5 operators with their counterparts: (1) *Inversion*: We use the standard Euler ODE solver to reverse the trajectory instead of DDIM inversion; (2) *Stochasticity*: We introduce a controlled noise term into the original ODE trajectories during the forward process, keeping the noise strength consistent with our Stable Diffusion 1.5 experiments.

**Evaluation Protocol.** The fine-tuning and evaluation strictly follow the protocol in the main paper. We utilize the standard Hugging Face `diffusers` scripts for FLUX.1 and Stable Diffusion v3. For each model, we perform LoRA fine-tuning on the DreamBooth dataset for 100 epochs, while maintaining all other hyperparameters at their default values to ensure a controlled environment.

**Results and Discussion.** As shown in Table 11, RECOVER achieves significant improvements in both ACC and AUC across these modern architectures compared to the T2I and I2I baselines. Notably, RECOVER's performance on Flow Matching models even surpasses its results on Stable Diffusion 1.5.

**Insight on Flow Matching.** We hypothesize that the superior performance on modern models is due to the "straighter" optimal transport trajectories inherent in Rectified Flow. These trajectories yield a cleaner signal during the inversion process, which in turn amplifies the *Inversion Robustness* gap between authorized and unauthorized data usage. This finding suggests that RECOVER is well-positioned to remain effective as generative backbone architectures continue to evolve.

*Table 11.* Detection performance on modern Flow Matching models (FLUX.1 (Labs, 2024) and Stable Diffusion v3 (Esser et al., 2024)). RECOVER significantly outperforms the T2I and I2I baselines.

| Model Architecture | Method | ACC (%) | AUC (%) |
|---|---|---|---|
| FLUX.1 | Text2Img | 67.72 | 69.20 |
| | Img2Img | 66.46 | 70.12 |
| | **RECOVER (Ours)** | **89.03** | **91.24** |
| Stable Diffusion v3 | Text2Img | 68.99 | 70.69 |
| | Img2Img | 69.62 | 72.55 |
| | **RECOVER (Ours)** | **94.30** | **96.93** |

## E.5. Potential Countermeasures

To comprehensively evaluate the robustness and generalizability of RECOVER, we conduct extensive experiments against four advanced adaptive attacks. All experiments are implemented based on the Stable Diffusion v1.5 text-to-image framework. For the evaluation dataset, we adopt the widely-used DreamBooth dataset, which covers 30 distinct categories to ensure a diverse and rigorous evaluation of concept fine-tuning detection.

The detailed experimental configurations and results for each adaptive attack are specified as follows:

### E.5.1. TRAINING-TIME AUGMENTATIONS

During the personalization fine-tuning phase, we introduce a random masking mechanism as a data augmentation technique to circumvent detection. Specifically, for each training image, a random binary mask with a missing ratio of $\tau$ is applied to obstruct a portion of the visual content, thereby attempting to disrupt the continuity of the model's learned denoising trajectories. In our implementation, we systematically vary the mask ratio $\tau$ from 0 to 0.8 with a step size of 0.2. To evaluate the consequences of this attack, we monitor both the detection accuracy (ACC) of RECOVER and the generation quality of the fine-tuned model, where the latter is quantified using the DINO score of the synthesized images.

As shown in Fig. 10, experimental results show that while both ACC and generation quality (DINO) decrease as $\tau$ increases, the ACC drop significantly lags behind the DINO decline. This indicates that RECOVER remains highly effective as long as the model successfully learns the target concept.

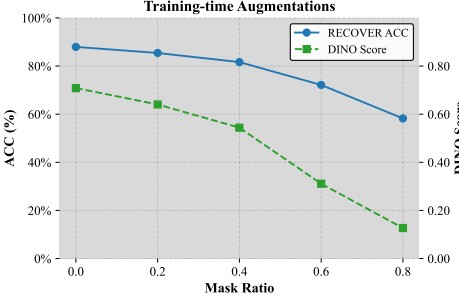

*Figure 10.* Results of the training-time augmentations experiment.

### E.5.2. TRAJECTORY SENSITIVITY REGULARIZATION

The Trajectory Sensitivity Regularization (TSR) attack aims to deliberately minimize the trace left by the fine-tuning process by penalizing the noise prediction shifts. The adversary incorporates an explicit regularization term into the standard diffusion loss function during training:

$$\mathcal{L}_{TSR} = -\lambda \|\epsilon_\theta(z_t + \delta, t, c) - \epsilon_\theta(z_t, t, c)\|_2^2 \qquad (20)$$

where $\delta$ represents a small random perturbation sampled from a Gaussian distribution to simulate potential trajectory shifts, and $\lambda$ is a balancing hyperparameter that controls the regularization strength. To thoroughly test our method against this defense, we evaluate the system under a wide spectrum of regularization scales, varying $\lambda$ across multiple orders of magnitude (e.g., from 0.2 to 1).

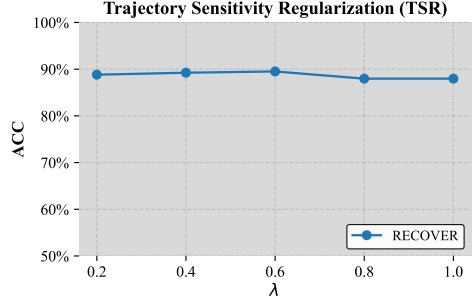

*Figure 11.* Results of the trajectory sensitivity regularization experiment.

*Table 12.* Adaptive attack experiments of RECOVER with respect to LoRA Rank ($r$) on Stable Diffusion v1.5, with other experimental settings identical to those in Tab. 1. The marginal performance fluctuations (5% in ACC) confirm that the local manifold shifts are robust across different parameter scales.

| LoRA Rank ($r$) | ACC | AUC |
|---|---|---|
| $r = 64$ | 89.87% | 94.23% |
| $r = 48$ | 88.61% | 92.37% |
| $r = 32$ | 87.97% | 92.01% |
| $r = 16$ | 86.08% | 90.51% |

*Table 13.* Comparison of time cost and memory usage across methods. RECOVER maintains both low time consumption and low VRAM usage.

| | No backward | Time costs (min)/Input Img | VRAM Usage |
|---|---|---|---|
| CGI-DM (Wu et al., 2024) | ✖ | 4.5 | 12GB |
| FineXtract (Wu et al., 2025) | ✔ | 0.8 | 8GB |
| Text2img | ✔ | 0.5 | 4GB |
| Img2img | ✔ | 0.5 | 6GB |
| RECOVER | ✔ | 0.5 | 4GB |

As shown in Fig. 11, experimental results reveals that regardless of how the regularization parameter $\lambda$ is adjusted, our method consistently achieves a high detection accuracy of 87.9%. This highlights RECOVER's strong robustness against trajectory-based regularization attacks.

### E.5.3. LoRA RANK SCALING

To evaluate whether the capacity of the fine-tuning parameter space affects the detection capability of RECOVER, we employ Low-Rank Adaptation (LoRA) to parameterize the concept fine-tuning. We inject LoRA layers into the Stable Diffusion U-Net architecture. During the evaluation, we systematically scale the intrinsic rank of the LoRA matrices from a high-capacity setting down to a heavily constrained low-rank setting, specifically exploring $r \in \{16, 32, 48, 64\}$.

As shown in Tab. 12, the evaluation across different fine-tuning scales shows that RECOVER's detection accuracy only fluctuates by a marginal 5% when scaling from rank 64 down to 16. This confirms the robustness of our detection mechanism across varying parameter capacities.

### E.5.4. TRIGGER PROMPT DISTORTION

To simulate real-world inference-time evasion where an attacker attempts to obfuscate the trigger words, we implement a Trigger Prompt Distortion attack. We leverage Qwen3.5 (Qwen Team, 2026) to automatically generate semantically equivalent but syntactically varied prompt mutations. For each original trigger prompt (e.g., "a [V] dog"), the LLM is instructed to produce diverse descriptive variations through paraphrasing, adopting alternative contexts, or inserting modifier words. RECOVER is then evaluated on these distorted prompts to verify if it can robustly detect unauthorized data usage.

When evaluated with the LLM-generated variant prompts, RECOVER successfully maintains an ACC of 87.75%, which represents merely a marginal 2% drop compared to the original prompts. This demonstrates its strong robustness against text-based evasion tactics during inference.

## F. Time and Resource Cost

In this section, we compare the time efficiency of RECOVER with existing non-intrusive detection approaches, demonstrating that RECOVER achieves high efficiency. We also compare it with image generation pipelines, showing that the comparison is fair in terms of runtime. As shown in Tab. 13, RECOVER is 9 × faster than CGI-DM and 1.6 × faster than FineXtract, demonstrating its high efficiency.

Since our method only requires a single model for inference, whereas CGI-DM involves gradient backpropagation and FineXtract requires simultaneous inference of two models, our GPU memory usage is the lowest.

# G. Detailed Discussion on the Threat Model

In the context of detecting unauthorized data usage in diffusion models, the threat model is defined by the level of access an auditor (e.g., a copyright holder or a regulatory body) has to the target model. Based on realistic commercial and legal constraints, we categorize the auditing environments into three distinct settings and discuss the applicability of RECOVERwithin them.

## G.1. White-Box Auditing

Existing high-performing detection baselines, such as CGI-DM (Wu et al., 2024), operate under a strict white-box assumption. They demand full access to the target model's internal weights and gradients to perform detection. In real-world commercial scenarios, this assumption is highly impractical. Generative AI companies consider their model weights as core intellectual property and strictly prohibit public access to prevent model theft or replication. Consequently, white-box methods face significant deployment hurdles in practical copyright disputes.

## G.2. Flexible Black-Box Auditing

Our proposed method, RECOVER, is designed for a much more practical *flexible black-box setting*. In this scenario, the auditor does not have access to model weights or gradients. Instead, the auditor only requires a forward-pass API that exposes fundamental generation parameters, specifically the scaling factor $\eta$.

We argue that this is the most realistic threat model for modern technical auditing for two main reasons:

- **Technical Availability:** To ensure high-quality and controllable generation for developers, mainstream commercial platforms (e.g., Replicate, Fal.ai) routinely expose parameters like $\eta$ (often termed as `etanumber`) to users.

- **Legal Compliance:** In real-world copyright litigation, recent legal cases demonstrate that courts can compel AI companies to cooperate with technical auditing mandates. Because RECOVER only requires forward-pass access with a commonly used parameter, companies can safely comply with these legal mandates without exposing their core IP (model weights or gradients). Thus, RECOVER provides an optimal, realistic trade-off between copyright auditing and model IP protection.

## G.3. Strictly Rigid Black-Box Auditing

A strictly rigid black-box setting is one where the target model only accepts text inputs and returns final images, abstracting away all internal generation parameters. We acknowledge that RECOVER cannot be directly applied in this restricted environment, as our method relies on controlling the forward $\eta$ to accurately estimate the inversion robustness. However, given the increasing legal demand for accountable AI and the technical necessity of generation control, the flexible black-box setting remains the dominant and most relevant scenario for future copyright protection frameworks.

# H. Visualizations

## H.1. Visualization for Inversion Robustness

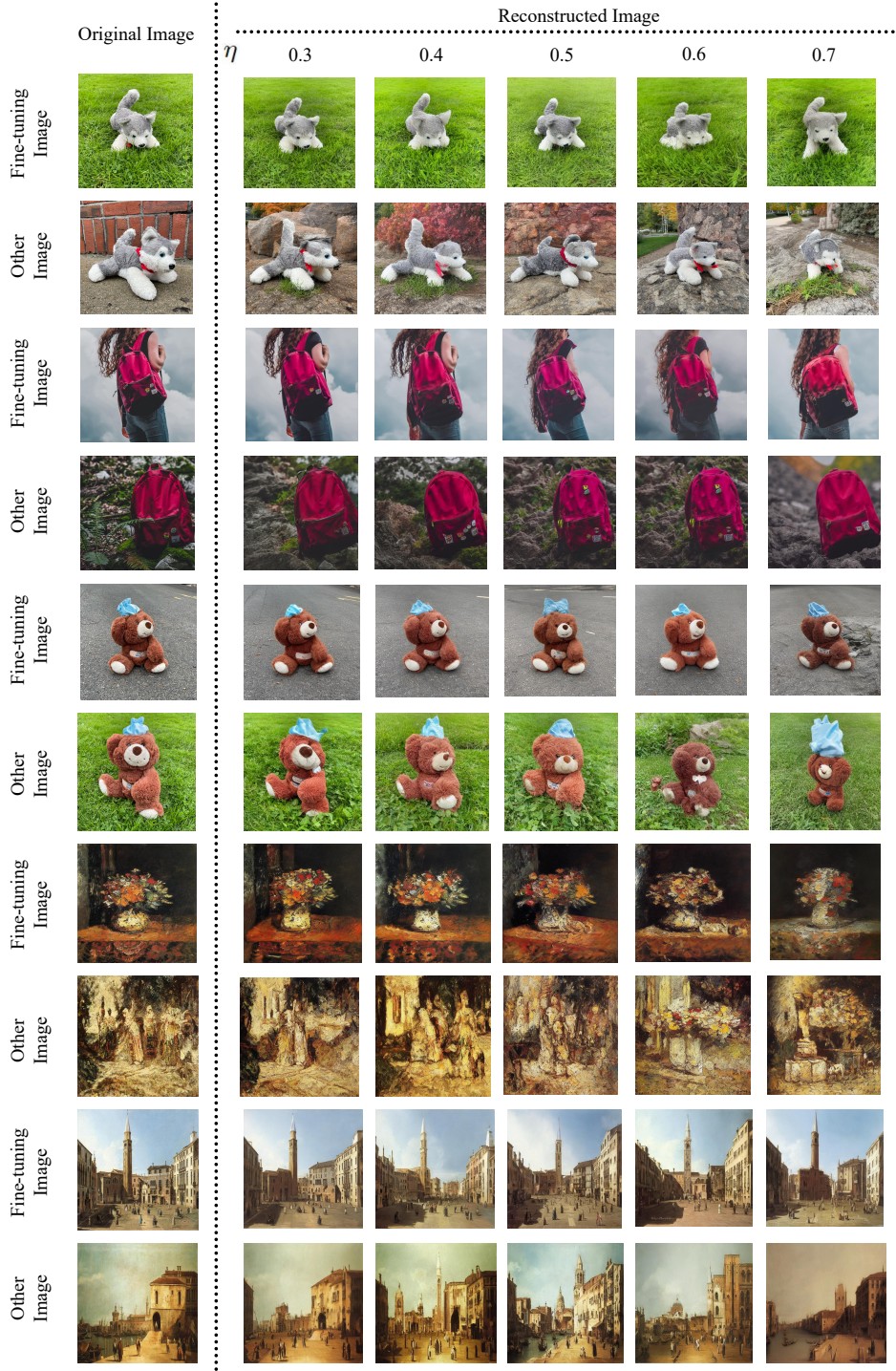

*Figure 12.* Additional visualization results illustrating inversion robustness.

## H.2. Visualization for Fine-tuning Steps

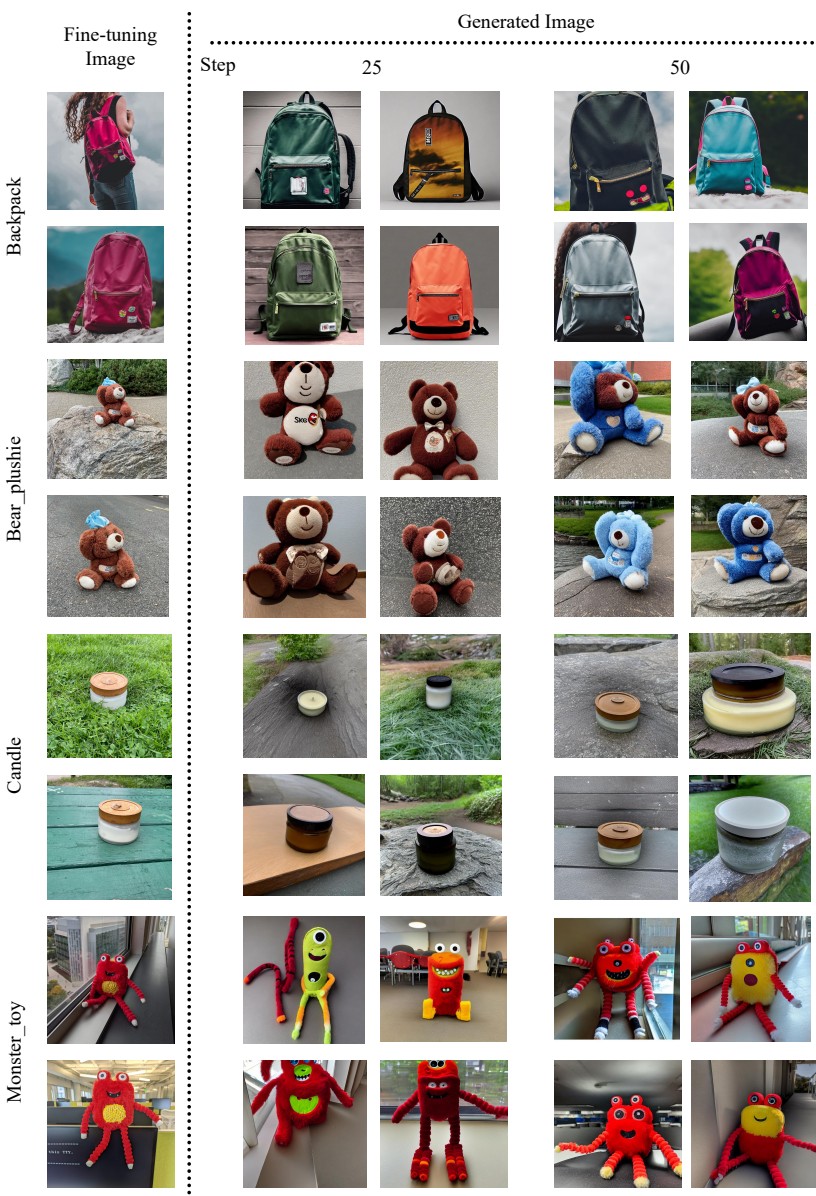

*Figure 13.* Visualization of direct text-to-image (T2I) generation results under low-step fine-tuning.

## H.3. Visualization for Limited Data Ratios

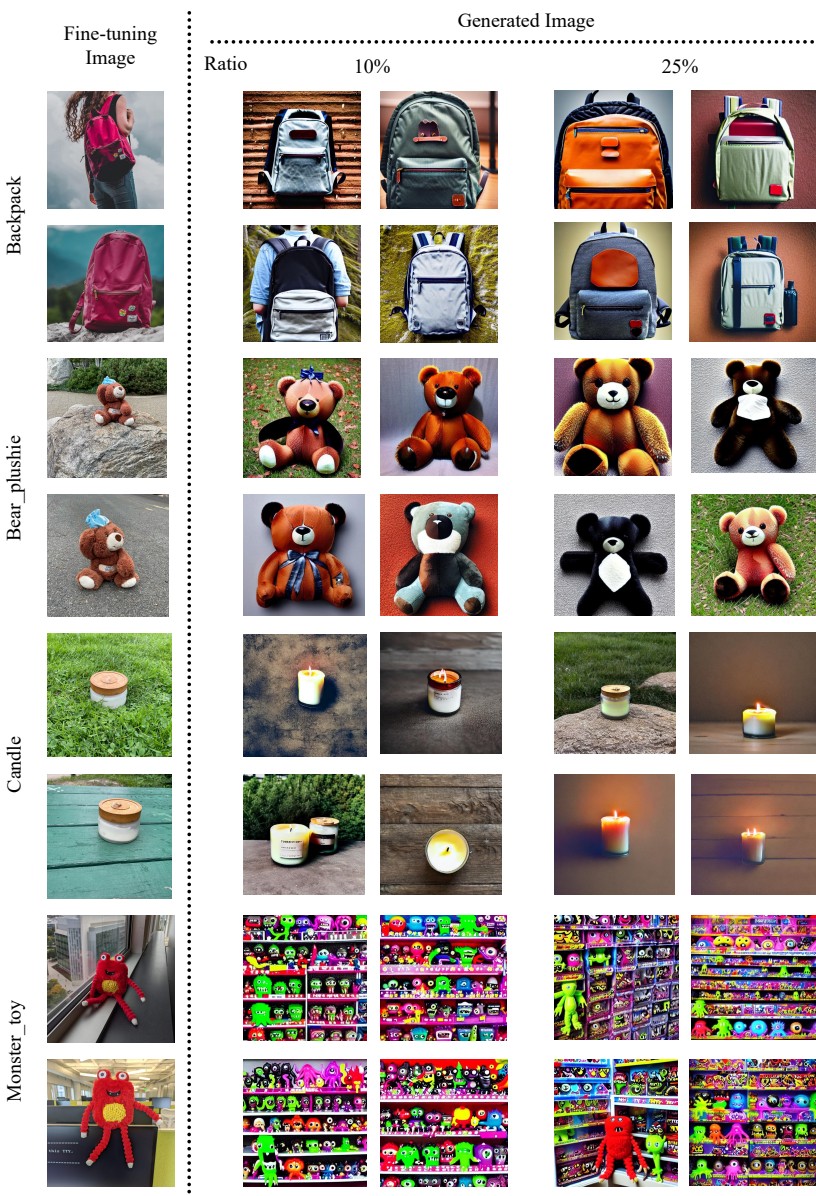

*Figure 14.* Visualization of direct text-to-image (T2I) generation results under low data ratio.

