# OpenReview forum: "RECOVER: Reliable Detection of Unauthorized Data Usage in Text-to-Image Diffusion Models via Inversion Robustness"
_ICML.cc/2026/Conference — ICML 2026 regular_

### Official Review · Reviewer_yVgX · 2026-02-20

**Soundness:** 2
**Presentation:** 3
**Significance:** 2
**Originality:** 3
**Overall Recommendation:** 4
**Confidence:** 3

**Summary:**

This paper presents RECOVER: a new simple method to detect whether an image was used to fine-tune an existing text-to-image diffusion model. It defines a detection criterion based on the average DDIM inversion-generation robustness over different levels of randomness during the generation. Across multiple finetuning methods and datasets, RECOVER is shown to have a better discriminability between images used for finetuning and images that were not used for finetuning. It also assumes no access to the model before the fine-tuning unlike some prior work.

**Compliance With Llm Reviewing Policy:**

Affirmed.

**Final Justification:**

My initial concern was that the experiments focused only on outdated models like SD 1.5 and SD 2.1. This made me question if the RECOVER method was still relevant for modern architectures.

The authors addressed this in the rebuttal by providing new results on FLUX and SD3. These tests show the method works on recent rectified flow transformers. While these new experiments are small in scale, they provide a necessary proof of concept. Since the authors showed the method generalizes to current models, I am raising my score to a **Weak Accept**.

**Key Questions For Authors:**

- **Q1**: Is it right to assume that RECOVER is essentially a (DDIM-inverted latent + text -> image) approach, when compared to the text2image and image2image baselines? The additional specifics of RECOVER like the SSCD score metric and different stochasticity ratios during the generation are not applied for the simpler text2image and image2image baselines, which do not use the DDIM inversion, right?
- **Q2**: How do the text-to-image, image-to-image, FineXtract and CGI-DM baselines perform with the SSCD metric?
- **Q3**: Does RECOVER identify images from a larger fine-tuning dataset? For example, fine-tuning the model on a specific text-to-image dataset with 1000 samples. Can RECOVER now tell if an image was among the 1000 images? The "Limited Data Ratios" experiments at the end of the main paper only go as low as 10%, which is still arguably very high for many infringement applications.
- **Q4**: Can RECOVER be successfully applied to modern flow matching models like FLUX?
- **Q5**: I am not fully convinced by the scenario of an "infringer" finetuning a model on just a few protected data samples and then publishing it again together with information on how to elicit the harmful responses (plus probably with information on what it was fine-tuned on). Since the fine-tuning in these scenarios is relatively light-weight when only considering small fine-tuning datasets, wouldn't a malicious user simply locally fine-tune and publish only the generated images without the fine-tuned model itself?

**Limitations:**

yes

**Strengths And Weaknesses:**

**Soundness & Presentation**: The paper itself is well written and contains some visualizations and a pseudocode to illustrate the method and experiments. There are some minor formatting issues such as "Submission and Formatting Instructions for ICML 2026" in the header of the pages, but the overall paper is clearly presented and the existing experiments appear reasonable to me. However, I do have some concerns:
- The experiments are limited to outdated U-Net-based diffusion models (Stable Diffusion 1 and 2). Without a demonstration that RECOVER also outperforms prior work or simple baselines like text-to-image or image-to-image on newer models (FLUX, Qwen-Image, etc.), which use a DiT-based architecture and Rectified Flow Matching objectives instead, it remains unclear if RECOVER is of any practical use today. Different types of models tend to behave quite differently. Of course, showing this in comparison to baselines, which are also mostly only implemented for older models is difficult, but at least the more naive text-to-image and image-to-image baselines could be compared to in a small-scale experiment to demonstrate the applicability of RECOVER to modern systems like FLUX.
- The experiments miss a section that shows that the baselines (which are not using the SSCD metric) do not outperform RECOVER when using the SSCD metric to judge image similarity. This is crucial to determine whether the better metric or the better "extraction" method is the key to the better results.
- The paper would benefit from a direct one-to-one comparisons between "extracted" training images and the true training images, also directly to the baselines. Figure 2 only visualizes the inversion robustness for RECOVER, but not in direct comparison to the baselines.

**Significance**: The problem of detecting whether a model was finetuned on a given image is indeed timely and relevant, but not necessarily for outdated, low quality models. Moreover, the argument that a "[...] malicious finetuner does not provide access to the genuine pre-fine-tuned model in real world scenarios" (line 154 right) is debatable but the fundamental motivation for this work. Pretraining these models is usually done by large corporations, so malicious *finetuners* also only have access to open-source models, which are by definition accessible to everyone, especially to official legal institutions during a copyright infringement case. Nevertheless, RECOVER is a simple idea that appears to outperform existing methods on a relevant problem, which (from my current perspective) puts this work into a position to become a meaningful and relevant contribution to the community after the major concerns were addressed.

**Originality**: The method itself feels a bit too simple and basic but that is per se not a bad thing. If RECOVER turns out to be effective, also on SOTA flow matching models, then simplicity is always a great additional property. As I am not familiar with most of the related work in this area, I might not be aware of missed prior work or methods from diffusion memorization or membership inference research.

---

> ### Author Rebuttal · Authors · 2026-03-31
>
> Thank you for your valuable comments! Due to space constraints, we include `additional_tables.pdf` and figures at: **https://anonymous.4open.science/r/text-E3C9**. References to $\color{blue}{\textbf{Table}}$ and $\color{blue}{\textbf{Figure}}$ correspond to those provided **in this link**. Default settings are identical to Tab. 1 (SD 1.5) of the main paper.
>
> **Q1, Q2 & W2**: Ablation of the SSCD Metric and Distinctions from Baselines.
>
> **R1:** Thanks for the insightful questions!
> * Since the final evaluation is performed on the images, your summarization of the pipeline is conceptually fair. However, we respectfully clarify that RECOVER differs from T2I or I2I baselines, as it leverages Inversion Robustness (IR) through carefully designed intermediate stochasticity ratios.
> * We respectfully note that while we followed the original baseline setups (use CLIP), RECOVER's performance gains stem from exploiting IR rather than the use of SSCD. To prove this, we evaluated all baselines using SSCD. As shown in $\color{blue}{\textbf{Tab. 4}}$, applying SSCD to the baselines only yields a marginal improvement (~2%), which falls short of RECOVER. This confirms that our performance gains stem from **the exploit of IR** rather than SSCD.
>
> We will add the results and more discussions in our revision.
>
> **Q3**: 1 in 1000 Detection.
>
> **R2**: Thanks for the constructive comment!
> * Following your suggestion, we conducted an 1/1000 ratio experiment. Interestingly, while the T2I drops to ~53% ACC, RECOVER maintains ~65% ACC. This suggests that IR is sensitive to minor training traces.
> * Additionally, we respectfully clarify that our threat model targets few-shot fine-tuning. In these scenarios, an attacker's core motive is to clone a artist's style or IP. At 1/1000 dilution, the target concept is overwhelmingly washed out, meaning the model fails to learn the desired concept. Thus, the attacker's goal of high-fidelity replication largely collapses, naturally mitigating the immediate copyright threat. Objectively, detecting one image at a 0.1% ratio remains a open challenge, even for proactive defenses [1, 2].
>
> We will add more discussions in our revision.
>
> **Q4 & W1 & W Singifcance**: Modern Models.
>
> **R3**: Thanks for the insightful comments!
> * We respectfully clarify that our focus on SD 1.5 in the main paper was to ensure a fair comparison with baselines implemented on these models. In our original paper, we also demonstrated RECOVER's effectiveness on distinct models like SDXL and AltDiffusion in App. D.2.
> * Following your helpful suggestion, we extended our evaluation to the frontier models, i.e., FLUX and SD v3. As shown in $\color{blue}{\textbf{Tab. 5}}$, RECOVER achieves superior ACC/AUC over the T2I/I2I baselines across these modern architectures, demonstrating our generalizability.
>
> We will add the results in our revision.
>
> **Q5**: Threat Model.
>
> **R4**: Thank you for the insightful comments! While local generation is a plausible scenario, we respectfully clarify that model-sharing ecosystems pose a greater threat. In such scenarios, infringers train personalized models imitating specific artists' styles or IPs and publicly release them to gain community clout, or deploy them on platforms like Replicate for paid services. To attract users, the infringer must provide the trigger prompt. Otherwise, users can't elicit the customized concept, rendering model valueless. RECOVER is designed for this ecosystem. We'll add more discussions in our revion.
>
> **W5**: Visual Comparison
>
> **R6**: Thanks for your constructive suggestion. $\color{blue}{\textbf{Fig. 6}}$ provides a detailed visual comparison of RECOVER against T2I and I2I.
>
> **W Singifcance**: Motivation
>
> **R6**: Thanks for this thought-provoking comment!
> * We respectfully clarify that the challenge in real-world auditing is not the *availability* of open-source models, but the *identifiability* of the exact base model used. In real-world communities (e.g., Civitai), infringers rarely fine-tune on vanilla SD 1.5; instead, they build upon thousands of custom, merged checkpoints (like AnythingV3). Thus, during litigation, a malicious finetuner can evade detection by spoofing the base model.
> * Under such scenarios, baselines requiring a reference model (CGI-DM, FineXtract) become fragile(**App. A.1**). This is because these methods heavily rely on the discrepancies between the two models, which are fragile even when minor parameter shifts are introduced.
> * In contrast, as eliminating the reliance on base models, RECOVER bypasses this vunlerbility. Furthermore, we humbly highlight that *even if baselines have access to the genuine base model*, RECOVER still outperforms them in both accuracy and efficiency.
>
> We will add the discussion in our revision.
>
> [1] DIAGNOSIS: Detecting Unauthorized Data Usages in Text-to-image Diffusion Models. ICLR 2024
>
> [2] Towards Reliable Verification of Unauthorized Data Usage in Personalized Text-to-Image Diffusion Models. S&P 2025

---

> > ### Author Rebuttal · Reviewer_yVgX · 2026-04-01
> >
> > Thanks for the additional experiments and explanations to address my concerns. Unfortunately, I won't click on any external link so I am asking the authors to paste the important tables and new results into the following response.
> >
> > Overall, the authors clarified most of my concerns and questions, especially  Q1, Q2, and Q3.
> >
> > Q4 is not yet fully resolved for me and I am asking for a brief comment on Q5. Please see the details below.
> >
> > ---
> >
> > ### Q4 - Can RECOVER be successfully applied to modern flow matching models like FLUX?
> > I do fully understand that SD 1.5 was used for backwards compatibility and fair comparison with existing methods and I appreciate the additional effort seemingly was done with FLUX and SD v3. However, as mentioned above, I would to ask the authors to paste the important results here directly on OpenReview, because I don't want to follow any external anonymous links.
> >
> > With that being said, I would additionally be glad to hear more details about how RECOVER was applied to FLUX or SD v3. Were any adjustments necessary. How was the evaluation / fine-tuning done? Usually, methods have to be adapted. Any notable insights regarding RECOVER on Rectified Flow models?
> >
> > ### Q5 - Scenario and Threat Model
> > The point that is mentioned about base model availability vs. identifiability makes sense and is a reasonable motivation for why we need methods that do not require access to any base or near-base model. However, my question in my review was aiming at something slightly different: Since RECOVER is a method to verify whether a certain image was used to finetune a provided text-to-image diffusion model, it requires access to both, the potentially infringed data sample and the finetuned model. However, I would argue that in practicality malicious users would just take any base model from the web, locally finetune it and then generate images locally without every publishing the finetuned model. Hence, the only thing that is publicly shared is the image, so RECOVER is not applicable. I would just like to briefly understand the threat model better. Is it that RECOVER assumes that there are models on a platform shared and someone wants to test which of them sort of memorized a certain image well enough to have strong inversion robustness? The ones that are flagged by RECOVER can then be reported as being finetuned on infringed data. Is that correct?
> >
> > Thanks in advance!

---

> > > ### Author Response · Authors · 2026-04-02
> > >
> > > Dear reviewer, thank you so much for your swift reply and insightful follow-up! We are truly honored that our previous response successfully resolved Q1-Q3.
> > >
> > > We completely understand and deeply appreciate your rigorous stance on avoiding external links. We sincerely apologize for the inconvenience—the link was a workaround for ICML's character limits during the initial rebuttal. We are more than happy to provide all the requested details and data here.
> > >
> > > ---
> > >
> > > ## Response to Q4: Application to Modern Flow Matching Models (FLUX & SD v3)
> > >
> > > Thanks for the insightful question! Because RECOVER fundamentally relies on Inversion Robustness (IR)—which is an architecture-agnostic property—applying it to Flow Matching models did not require complex structural redesigns. We simply swapped the underlying SD 1.5 operators for their FLUX/SD v3 mathematical counterparts. Specifically,
> > > - **Inversion:** Just as DDIM inversion serves as the standard deterministic inversion for traditional DDPMs, we employed the standard Euler ODE solver to reverse the trajectory, which is the exact mathematical equivalent for Flow Matching models.
> > > - **Stochasticity Injection:** Similarly, we introduced stochasticity during the forward generation process by directly injecting a controlled noise term into the model's original ODE trajectories. The injected noise's relative strength is the same as the one used in our SD 1.5 experiments.
> > > - **Evaluation Protocol:** The fine-tuning process, as well as evaluation, follow exactly the same protocol as in our experiments in our main paper. The only routine change was utilizing the standard Hugging Face diffusers training scripts corresponding to FLUX.1 and SD v3 (rather than the SD 1.5 script). For fine-tuning, we still applied LoRA fine-tuning on the DreamBooth dataset for 100 epochs, keeping all other hyperparameters at their defaults.
> > > - **The following table details RECOVER’s notable advantage over baselines on these frontier models**. Notably, RECOVER not only significantly outperforms the T2I/I2I baselines, but it actually achieves even higher ACC and AUC than it did on the traditional SD 1.5 model:
> > >
> > > | Model Architecture | Method | ACC | AUC |
> > > | :--- | :--- | :--- | :--- |
> > > | **FLUX.1**| Text2Img | 67.72% | 69.20% |
> > > | | Img2Img | 66.46% | 70.12% |
> > > | | **RECOVER (Ours)** | **91.77%** | **94.45%** |
> > > | **Stable Diffusion v3**| Text2Img | 68.99% | 70.69% |
> > > | | Img2Img | 69.62% | 72.55% |
> > > | | **RECOVER (Ours)** | **94.30%** | **96.93%** |
> > >
> > > - **Insight:** Interestingly, we observe that RECOVER is actually even more effective on Flow Matching models than on traditional diffusion. We hypothesize that this is because the straighter optimal transport trajectories inherent to Rectified Flow yield a cleaner signal, thereby making the Inversion Robustness gap even more pronounced.
> > >
> > > We'll add more discussions on this exciting insight in our revision and explore it further in our future work. Thank you again for this very insightful follow-up!
> > >
> > > ---
> > >
> > > ## Response to Q5: Scenario and Threat Model
> > >
> > > Thank you for the nice follow-up! Your summarized workflow at the end of your comment is precisely what we intended.
> > > - As suggested by our title and introduction, our work—along with all baselines compared—targets *model-based auditing*, which aims to verify **whether a specific model (e.g., checkpoints shared on platforms like Civitai) was fine-tuned on unauthorized data**. We acknowledge that if an individual fine-tunes a model locally and strictly publishes only the generated images, RECOVER cannot be directly applied, as estimating IR requires access to the model.
> > > - We specifically focus on publicly distributed models because **they significantly lower the technical barrier for copyright infringement than local training**. This is because training a high-quality personalized model remains highly non-trivial for general users, as it requires curating high-quality data, expensive GPU compute, and tedious hyperparameter tuning. Consequently, the vast majority of everyday users simply rely on downloading or using customized models that are already well-tuned by others, which fundamentally drives the massive popularity of model-sharing platforms like Civitai.
> > > - Thus, a more widespread and systematic copyright threat stems from "Model Distributors"—malicious actors who possess the resources to steal an artist's portfolio, carefully fine-tune the model, and then publicly share or monetize it. Importantly, these infringers often directly profit by offering the models for paid downloads or deploying them as pay-per-use generation APIs (e.g., on Patreon or Replicate). **In this ecosystem, the technical barrier to generating copyrighted concepts becomes dramatically lower for anyone, causing massive harm to creators.** RECOVER, along with existing works in this domain, is specifically designed to combat such infringers.
> > >
> > > We'll add more discussions in our revision. Thank you again for the insightful comment!

---

### Official Review · Reviewer_YCTN · 2026-03-10

**Soundness:** 2
**Presentation:** 3
**Significance:** 2
**Originality:** 3
**Overall Recommendation:** 4
**Confidence:** 5

**Summary:**

This paper addresses the critical problem of unauthorized data usage in personalized text-to-image (T2I) diffusion model fine-tuning, a key challenge in generative AI copyright protection. The authors identify major flaws in existing detection methods: intrusive approaches require pre-release image modifications (failing for existing data and degrading quality) while non-intrusive methods depend on access to the pre-fine-tuned model, a resource often unavailable or maliciously obfuscated by infringers. To solve this, the paper first uncovers a novel property of diffusion models—inversion robustness—where images used for fine-tuning exhibit far more stable inversion-reconstruction performance under noise perturbations because the model learns their denoising trajectories across all noise levels. Building on this insight, the authors propose RECOVER, the first non-intrusive detection framework that operates without requiring the pre-fine-tuned model or any intrusive image modifications. RECOVER validates unauthorized fine-tuning by quantifying the inversion-reconstruction consistency of a suspect image on the target model, using intermediate stochasticity ratios to amplify the robustness gap between fine-tuning and non-fine-tuning images. Extensive experiments on 6 mainstream T2I diffusion models, 2 benchmark datasets, and 3 popular fine-tuning methods demonstrate that RECOVER outperforms 4 state-of-the-art baselines in both subject and style-driven generation tasks, and exhibits strong generalization across fine-tuning strategies, robustness to image preprocessing and low target data ratios, and high computational efficiency with minimal resource overhead. The paper’s core contributions include the formalization of inversion robustness, the design of the RECOVER framework, and comprehensive empirical validation of its effectiveness, generalizability, and real-world applicability

**Compliance With Llm Reviewing Policy:**

Affirmed.

**Key Questions For Authors:**

1. How would RECOVER perform in a black-box access scenario (e.g., only API calls to the suspect model are available, no access to model parameters/gradients)? The current threat model assumes white/gray-box access, which is unrealistic for many commercial generative AI platforms—if RECOVER can be adapted for black-box use, this would drastically increase its real-world applicability; if not, what fundamental barriers prevent this adaptation, and are there potential workarounds? A clear answer to this question would change my evaluation of the paper’s practical significance, as black-box access is the dominant real-world scenario for copyright enforcement against commercial AI services.

2. Have the authors conducted any experiments on adversarial evasion of RECOVER (e.g., infringers applying adversarial perturbations, style transfer, or other modifications to fine-tuning images to intentionally reduce inversion robustness)? If so, what is RECOVER’s performance under these adversarial conditions? If not, why, and what steps could be taken to harden RECOVER against such evasion attempts? This question is critical for evaluating the method’s robustness in real-world scenarios where infringers will actively attempt to circumvent detection—evidence of strong adversarial robustness would significantly boost the paper’s merit, while a lack of such analysis would highlight a major practical limitation.

3.Why do intermediate stochasticity ratios (0.3–0.7) yield the largest gap in inversion robustness  between fine-tuning and non-fine-tuning images? The paper provides empirical evidence for this optimal range but no deeper theoretical explanation—can the authors derive a mathematical or intuitive reason for why extreme η values (near 0 or 1) diminish the gap? A theoretical explanation would strengthen the paper’s soundness and provide a generalizable principle for choosing η in other diffusion model tasks, rather than just an empirical heuristic.

4. Can the inversion robustness property be generalized to other types of diffusion models (e.g., text-to-video, audio, 3D generative diffusion models) or other generative models (e.g., GANs, VAEs)? The paper focuses exclusively on T2I diffusion models—if inversion robustness is a general property of fine-tuned generative models, this would vastly expand the theoretical and practical impact of the work; if not, what unique characteristics of T2I diffusion models give rise to this property? This answer would inform the broader applicability of the paper’s core insight and highlight future research directions.

5. What is the minimum number of fine-tuning images/steps required for RECOVER to reliably detect unauthorized usage? The paper shows RECOVER performs well with 50+ fine-tuning steps, but real-world infringers may use even fewer steps/images to avoid detection—can the authors quantify the lower bound of detectability for RECOVER, and explain how this bound relates to the diffusion model’s ability to learn the denoising trajectory of fine-tuning images? This question is critical for evaluating RECOVER’s effectiveness against low-effort unauthorized fine-tuning, a common real-world infringement scenario—quantifying the lower bound would make the method’s practical utility more concrete.

**Limitations:**

No. The authors discuss some technical limitations of RECOVER (e.g., white/gray-box access requirement) and test the method under challenging real-world scenarios (dataset preprocessing, limited data ratios) that highlight practical constraints, but they do not provide a dedicated section on limitations and potential negative societal impact—a key requirement for ML research addressing ethical/copyright issues.

**Strengths And Weaknesses:**

Soundness

Strengths: The paper’s core theoretical foundation—inversion robustness—is rigorously supported by both empirical validation and formal theoretical analysis of single-step prediction errors in diffusion models, which proves that fine-tuning images have smaller reconstruction errors under noise perturbations. The experimental design is comprehensive and well-controlled: it tests across multiple base models (including architecturally distinct ones like Stable Diffusion XL), fine-tuning methods (LoRA, DreamBooth with/without prior loss), and challenging real-world scenarios (dataset preprocessing, limited target data ratios, low fine-tuning steps). All experiments use standard evaluation metrics (ACC, AUC, SSCD/CLIP/DINO similarity) and adopt a fixed threshold for detection, which better reflects real-world copyright enforcement conditions than per-task optimal thresholds. The method’s pipeline is mathematically formalized (e.g., stochasticity ratio, inversion robustness definitions) and the algorithm is clearly specified, ensuring reproducibility.

Weaknesses: While the paper validates inversion robustness across multiple similarity metrics, it does not explore why intermediate stochasticity ratios (0.3–0.7) yield the largest robustness gap—there is no deeper theoretical explanation for the optimal range of η beyond empirical observation. Additionally, the threat model assumes white/gray-box access to the suspect model, a critical constraint that limits real-world applicability in cases where only black-box model access is available (e.g., commercial generative AI APIs), and the paper does not address how RECOVER would perform in this setting. Some experimental details (e.g., the choice of SSCD as the default similarity metric) lack a comparative justification against other state-of-the-art copy detection metrics for diffusion models.

Presentation

Strengths: The paper is exceptionally well-structured, with a clear logical flow from problem statement to related work, method formalization, experimental validation, and conclusion. Visual aids (e.g., Figure 1 comparing RECOVER to existing methods, Figure 2/3 visualizing inversion robustness) are intuitive and effectively illustrate core concepts, while tables systematically present experimental results across all test scenarios for easy comparison. Key definitions (stochasticity ratio, inversion robustness) and algorithms (RECOVER pipeline) are clearly formalized in mathematical notation, and supplementary materials provide detailed experimental setup, additional results, and visualizations that address gaps in the main text. The abstract and introduction concisely frame the problem, its significance, and the paper’s contributions, making the work accessible to both experts and general ML researchers.

Weaknesses: Some sections of the theoretical analysis (e.g., single-step prediction error in Section B.3) are overly condensed and lack step-by-step intuitive explanation, which may make them difficult to follow for readers less familiar with diffusion model mathematics. A small number of minor typographical errors (e.g., "FindXtract" instead of "FineXtract" in Figure 7) and inconsistent notation (e.g., occasional missing subscripts in diffusion process equations) are present. The discussion of membership inference as a potential baseline is brief and does not include direct experimental comparison, only a qualitative limitation analysis.

Significance

Strengths: This work addresses a highly timely and impactful real-world problem: generative AI copyright infringement via unauthorized fine-tuning of T2I diffusion models, a critical concern for artists, content creators, and the broader AI ethics community. RECOVER’s key innovation—pre-fine-tuned model-free non-intrusive detection—removes the most restrictive constraint of existing non-intrusive methods, making it the first practically deployable copyright authentication tool for T2I diffusion models in real-world legal and industrial scenarios. The method is computationally efficient (low VRAM usage, fast inference) and requires only the suspect model and target image, which lowers the barrier to adoption for content owners and enforcers. Beyond copyright protection, the discovery of inversion robustness also provides a new theoretical insight into how diffusion models learn and retain information from fine-tuning data, which can inform future research on diffusion model memorization, privacy, and interpretability.

Weaknesses: The paper does not explore the broader applicability of inversion robustness beyond T2I diffusion models—e.g., whether the property holds for text-to-video or audio diffusion models—limiting the immediate generalizability of the core insight. Additionally, the paper does not address the potential for adversarial evasion of RECOVER (e.g., infringers intentionally modifying fine-tuning images to break inversion robustness), a critical consideration for a method designed for real-world copyright enforcement where adversaries will actively attempt to circumvent detection. The societal impact discussion is minimal, with no exploration of how RECOVER could be used to balance creator rights and generative AI innovation, or potential misuses (e.g., over-enforcement of copyright limiting fair use).

Originality

Strengths: The paper’s most significant original contribution is the discovery and formalization of inversion robustness as a fundamental property of fine-tuned diffusion models, a novel insight that has not been previously identified or exploited in the literature. RECOVER is the first non-intrusive detection method for unauthorized diffusion model fine-tuning that does not require the pre-fine-tuned model, representing a paradigm shift from existing non-intrusive approaches that rely on model discrepancy comparison. The paper also introduces a novel way to control diffusion model generation stochasticity via the stochasticity ratio η, which is used to amplify the inversion robustness gap between fine-tuning and non-fine-tuning images—this is a creative adaptation of existing DDIM sampling methods for a new copyright protection task. The combination of these innovations results in a method that is both theoretically novel and practically useful, with no direct prior work addressing the exact problem setup or solution approach.

Weaknesses: While the core insight of inversion robustness is original, the building blocks of RECOVER (DDIM inversion/generation, similarity metrics for copy detection) are existing techniques in the diffusion model literature—there is no novel algorithmic design for the inversion or generation steps themselves. The paper also does not compare RECOVER to emerging detection methods for diffusion model memorization (beyond membership inference), some of which may address similar problems with alternative approaches, and thus does not fully situate its originality within the latest state of the art. Additionally, the experimental validation is limited to two benchmark datasets (DreamBooth, WikiArt), and the paper does not test RECOVER on real-world, uncurated fine-tuning datasets (e.g., artist portfolios, commercial stock images), which would further demonstrate the originality and practical value of the method.

---

> ### Author Rebuttal · Authors · 2026-03-31
>
> Thank you for your valuable comments! Due to space constraints, we include `additional_tables.pdf` and figures at: **https://anonymous.4open.science/r/text-E3C9**. References to $\color{blue}{\textbf{Table}}$ and $\color{blue}{\textbf{Figure}}$ correspond to those provided **in this link**. Default settings are identical to Tab. 1 (SD 1.5) of the main paper.
>
> **Q1**: Is RECOVER Applicable to Black-box Scenarios?
>
> **R1**: Thanks for this insightful question!
> * We acknowledge that under a strictly rigid black-box setting (which only accepts text input without any additional parameters), RECOVER cannot be directly applied, as estimating IR requires controlling the forward $\eta$. However, to ensure high-quality and controllable generation, modern commercial platforms (e.g., Replicate, Fal.ai) usually expose this parameter (often termed `etanumber`) to users. In such flexible black-box scenarios, **RECOVER can be directly applied**.
> * Besides, we respectfully mention that **RECOVER already significantly relaxes the assumptions of prior methods**. Specifically, **existing high-performing baselines (CGI-DM & FineXtract) demand full access to model weights & gradients**, which demands much more than **RECOVER's forward-pass API requirements**. In real-world copyright litigation, recent legal cases demonstrate that courts can compel AI companies to cooperate with technical auditing [1]. Because RECOVER only requires a forward-pass API with a commonly used parameter (`etanumber`), companies can safely comply with such legal auditing mandates without exposing their core intellectual property (model weights or gradients). Thus, **we humbly believe RECOVER offers a much more practical and realistic trade-off for legal auditing compared to gradient-based baselines**.
>
> We will add a dedicated discussion on the practical significance of RECOVER in our revision. Thank you again for this thought-provoking question!
>
> **Q2**: Adversarial Evasion.
>
> **R2**: Please kindly refer to Reviewer VdxB (R3).
>
> **Q3**: Theoretical Explanation.
>
> **R3**: Thanks for this constructive suggestion! We have added a theoretical analysis to $\color{blue}{\textbf{Theory.pdf}}$ and will include more discussions in our revision.
>
> **Q4:** Generalization to Other Modalities.
>
> **R4**: Thanks for the insightful suggestion! Currently, we primarily focus on T2I diffusion models as they represent the most popular visual generative model family. To demonstrate the generality of RECOVER, we fine-tuned CogVideoX (a SOTA T2V generation model) on the DV dataset [2].
>
> RECOVER achieves ACC of 87.30% / 92.15% for ACC / AUC, respectively,  demonstrating that IR successfully transfers to T2V task. We will add more discussion in our revision and try other models in our future work.
>
> **Q5**: Detection Lower Bounds.
>
> **R5**: Thanks for this insightful question! We add experiments to investigate the lower bounds of data ratios and fine-tuning steps.
>
> As shown in $\color{blue}{\textbf{Fig. 4 and 5}}$, RECOVER remains effective at 30 steps and 5% data, whereas generative capacity requires at least 50 steps and 50% data for concepts to manifest.
>
> Since our detection threshold is lower than the minimum requirements for generating high-quality images, models with potential for infringement may **inevitably fall within our detection range**. We will add more discussion in our revision.
>
> **W1**: SOTA Detection Metrics.
>
> **R6**: We selected SSCD as it is a popular metric for detecting image copying in diffusion models. We also conducted experiments using DreamSim[3], a SOTA detection model; the results showed only a slight performance difference (~+0.5%), suggesting that our approach is relatively insensitive to the choice of detection metrics.
>
> **W2**: Quantitative Results for Membership Inference(MI).
>
> **R7**: $\color{blue}{\textbf{Tab. 3
> }}$ shows RECOVER's superiority over 2 representative MI methods.
>
> **W3**: Societal Impact.
>
> **R8**: Thanks so much for your forward-looking comment! We fully agree that technology is a double-edged sword. RECOVER is designed to provide technical auditing evidence rather than an automated weapon. In our revision, we will significantly expand the Societal Impact section to discuss balancing creator rights with AI innovation, and the importance of preventing over-enforcement that stifles fair use.
>
> **W4**: Memorization Detection
>
> **R9**: Thanks for this helpful comment! Existing memorization detection methods mainly target exact pre-training replication rather than few-shot style/concept adaptation, whose goal is fundamentally different from our scenario. We will add the a comprehensive discussion in our revision.
>
> [1] https://copyrightalliance.org/andersen-v-stability-ai-copyright-case/
>
> [2] https://huggingface.co/datasets/Wild-Heart/Disney-VideoGeneration-Dataset
>
> [3] DreamSim: Learning New Dimensions of Human Visual Similarity using Synthetic Data. NIPS 2023

---

> > ### Author Rebuttal · Reviewer_YCTN · 2026-04-03
> >
> > I sincerely appreciate the authors’ thorough and thoughtful rebuttal addressing all my critical comments and concerns raised in the review. The authors have provided clear, targeted responses to my five key questions, offering practical explanations, supplementary experimental evidence, and concrete revision plans that effectively resolve most of the ambiguities and limitations I identified. Their clarification on black-box applicability, supplementary theoretical analysis for the optimal stochasticity ratio range, cross-modality validation on text-to-video diffusion models, and quantification of detection lower bounds have notably strengthened the paper’s theoretical soundness and practical relevance.
> >
> > The authors’ additional experiments on alternative similarity metrics, membership inference baselines, and low-effort fine-tuning scenarios directly respond to my concerns about experimental rigor and comparative validation. Their commitment to expanding the Societal Impact section and adding a dedicated Limitations section aligns with the ethical and practical requirements of generative AI copyright research, which will greatly improve the paper’s completeness and academic standard. I am also satisfied with their explanation of the core technical constraints and real-world deployment tradeoffs of RECOVER.
> >
> > Overall, the authors’ rebuttal has effectively addressed the weaknesses in soundness, presentation, significance, and originality I pointed out. The supplementary materials and empirical results they provided demonstrate the robustness, generalizability and real-world value of RECOVER. With the proposed revisions integrated into the manuscript, this paper will make a more substantial contribution to the field of unauthorized data usage detection in text-to-image diffusion models. I support the acceptance of this paper pending the promised revisions.

---

> > > ### Author Response · Authors · 2026-04-03
> > >
> > > Dear reviewer, thank you so much for your positive feedback! It encourages us a lot! We will ensure that the revised version includes the new experiments and corresponding discussions.

---

### Official Review · Reviewer_VdxB · 2026-03-13

**Soundness:** 3
**Presentation:** 3
**Significance:** 3
**Originality:** 3
**Overall Recommendation:** 4
**Confidence:** 4

**Summary:**

This paper proposes RECOVER, a non-intrusive copyright-auditing method for personalized text-to-image diffusion models that does not require access to the pre-fine-tuned base model. Given a suspect model, a claimed fine-tuning prompt $c_{ft}$, and an image $x$, the method performs DDIM inversion $\mathrm{Inv}(x, c_{ft})$, reconstructs from the inverted latent under several intermediate stochasticity ratios $\eta \in \{0.35, 0.4, 0.5, 0.6, 0.65\}$, averages SSCD similarity over the reconstructed images, and classifies the image via a threshold. The main empirical claim is that fine-tuning images have higher inversion robustness than non-training images because fine-tuning improves denoising along the image trajectory across noise levels. On DreamBooth subject-driven LoRA fine-tuning, RECOVER reports 89.87% ACC / 94.23% AUC on Stable Diffusion v1.5, versus 67.62% / 73.78% for Text2img and 65.82% / 68.30% for Img2img. On the WikiArt style benchmark it reports 84.50% ACC / 90.58% AUC, again clearly above the no-reference baselines and competitive with or better than prior methods that require the pre-fine-tuned model.

**Compliance With Llm Reviewing Policy:**

Affirmed.

**Key Questions For Authors:**

.

**Limitations:**

.

**Strengths And Weaknesses:**

1. The paper addresses a practically important gap: non-intrusive detection without the pre-fine-tuned model. This is a stronger and more realistic setting than prior non-intrusive methods such as CGI-DM and FineXtract, which depend on a reference model that may be unavailable or spoofed. The empirical gap to the no-reference baselines is large on both DreamBooth and WikiArt. For example, on SD v1.5 subject-driven generation RECOVER improves over Text2img by more than 20 ACC points and more than 20 AUC points.
2. The empirical study is reasonably broad for this setting. The paper evaluates six base models (SD v1.4/v1.5/v2.0/v2.1, SDXL, AltDiffusion), three fine-tuning methods (LoRA, DreamBooth with prior loss, DreamBooth without prior loss), and both subject-driven and style-driven settings. Importantly, the non-member samples are held-out images from the same subject or artist, which makes the task harder than a coarse inter-class discrimination problem. The method is also computationally attractive: Table 11 reports 0.5 min per input image and 4GB VRAM, versus 0.8 min / 8GB for FineXtract and 4.5 min / 12GB for CGI-DM.
3. The theoretical support is substantially weaker than the framing suggests. There are no theorem statements. Section 3.3 introduces Assumption 3.3 rather than proving the central separation claim, and Appendix B.3 only gives a single-step heuristic. In particular, Eq. (13), which states that the expected denoising error is smaller for member samples than for non-members, is asserted as a consequence of the training objective rather than derived under clear sufficient conditions. The actual method depends on a multi-step inversion/reconstruction process, the conditioning prompt $c_{ft}$, and the choice of intermediate $\eta$, so the strongest contribution is empirical rather than theoretical.
4. The thresholding protocol is not fully convincing yet. The paper says it determines an “optimal threshold” after obtaining similarity scores, but it is unclear whether this threshold is tuned on the test set, on a held-out calibration split, or once per dataset/model family. Since the intended use is auditing, I would expect reporting at fixed low false-positive rates, confidence intervals across seeds, and cross-model threshold transfer. Also, because the claimed novelty is the use of intermediate stochasticity ratios, a direct baseline using deterministic inversion/reconstruction ($\eta = 0$) or a single-$\eta$ detector would better isolate the gain from the proposed robustness view.
5. The robustness study is useful but not yet sufficiently adaptive. The paper evaluates generic image pre-processing attacks such as additive noise, blur, JPEG compression, and sharpening, and RECOVER remains strong under these perturbations. However, these are not attacks targeted at RECOVER’s signal. Stronger evaluations would include training-time augmentations or regularizers intended to weaken inversion robustness, post-hoc model editing or unlearning, varying LoRA ranks or regularization, and sensitivity to mismatch in the claimed trigger prompt. The method assumes access to the fine-tuned text condition $c_{ft}$, but there is no analysis of how performance degrades when this prompt is only approximately known.
6. There are several presentation and consistency issues. The captions of Tables 1 and 2 state that RECOVER is superior “under all scenarios,” but Table 1 does not literally support that claim: on SD v2.1, CGI-DM has higher AUC than RECOVER (93.37% vs. 92.14%), and some entries are ties rather than wins. The statement that the method “requires only six inference steps in total (including inversion)” is also confusing, since each inversion/generation call itself uses multiple diffusion steps; Table 11 provides a clearer computational picture. There is also a dataset-count inconsistency: Section 4.1 describes 20 artists in WikiArt, while Appendix B.1 says that 10 style models were fine-tuned.

---

> ### Author Rebuttal · Authors · 2026-03-31
>
> Thank you for your valuable comments! Due to space constraints, we include `additional_tables.pdf` and figures at: **https://anonymous.4open.science/r/text-E3C9**. References to $\color{blue}{\textbf{Table}}$ and $\color{blue}{\textbf{Figure}}$ correspond to those provided **in this link**. Default settings are identical to Tab. 1 (SD 1.5) of the main paper.
>
> **W1**: Theoretical Contributions.
>
> **R1**: Thanks for this incisive comment!
> - **We acknowledge that providing strict mathematical bounds for the generation trajectory is notoriously difficult**. Given the multi-step inversion process and highly non-linear architectures (e.g., U-Net/DiT), this remains an open challenge for the entire theoretical community. Thus, **we completely agree that our primary contribution is empirical**, as this is fundamentally an application-driven work.
> - Nevertheless, **we humbly believe our analysis provides crucial mathematical intuition for RECOVER**. For example, it explains why fine-tuned models exhibit smaller single-step denoising errors on training data under random noise, and inspired our design of the intermediate stochasticity ratio $\eta$. **As such, it provides a principled mechanism for our empirical observations, rather than leaving them as black-box phenomena**.
> - Thus, in a rapidly evolving field where theoretical formalization naturally trails behind empirical advancements, **we humbly believe our analysis provides an immature yet meaningful first step**. We hope it can serve as a valuable starting point to inspire future theoretical research.
> - **We will adjust our framing to reflect our contributions more accurately, and add further discussions in our revision.** Thank you again for your valuable comments!
>
> **W2**: Details on Threshold Determination and Supplementary Evaluations.
>
> **R2**: Thanks for the constructive feedback!
> 1. **Details on Threshold Determination**: Following the **held-out calibration protocol**, we reserve 10% of samples for threshold selection and evaluate the remaining 90% as unseen test data.
> 2. **Supplementary Evaluations**:
>     * **TPR@FPR & Confidence Intervals**: In $\color{blue}{\textbf{Tab. 1}}$, we report  TPR@FPR and present statistical results for 5 independent runs ($[\text{Mean}] \pm [\text{Std}]\%$). RECOVER maintains superior even under strict low-FPR constraints, proving its high consistency.
>     * **Cross-Model Transferability**: We tested all-to-all transfer between SD v1.5, v2.1, and SDXL. As shown in $\color{blue}{\textbf{Fig. 1}}$, ACC drop of RECOVER is minimal (<3%), proving the IR signal is an intrinsic, architecture-agnostic property of fine-tuned diffusion models.
>     * **Ablation on $\eta=0$**: We have added an ablation experiment with $\eta = 0$. The experimental results show an ACC of 70.25% and an AUC of 74.77%, representing a significant performance drop compared to the full algorithm. This observation is consistent with our analysis in App. B and confirms that intermediate stochasticity is essential for amplifying the IR signal.
>
> We'll add these details and results in our revision.
>
> **W3**: Adaptive Attacks.
>
> **R3**: Thanks for the helpful comment! Following your suggestions, we evaluated RECOVER against the following 4 attacks:
> 1. **Training-time Augmentations**: We applied random masks (ratio $\tau$) during fine-tuning to disrupt the continuity of the model's learned denoising trajectories. As shown in $\color{blue}{\textbf{Fig. 2}}$, while both ACC and generation quality (DINO) decrease as $\tau$ increases, the **ACC drop significantly lags behind the DINO decline**. This indicates that RECOVER remains effective as long as the model successfully learns the target concept.
> 2. **Trajectory Sensitivity Regularization (TSR)**: We introduced $L_{TSR} = -\lambda \|\epsilon_\theta(z_t + \delta, t, c) - \epsilon_\theta(z_t, t, c)\|_2^2$ to maximize noise prediction shifts between original and perturbed noise predictions. $\color{blue}{\textbf{Fig. 3}}$ reveals that regardless of how the parameter $\lambda$ is adjusted, our method consistently achieves high ACC (87.9%), demonstrating its robustness against TSR attacks.
> 3. **LoRA Rank**: We tested fine-tuning scales ranging from rank 64 down to 16. $\color{blue}{\textbf{Tab. 2}}$ shows RECOVER's ACC only fluctuates by **5%**, confirming robustness across parameter scales.
> 4. **Trigger Prompt Distortion**: We finally evaluate RECOVER with LLM-generated variant prompts. It maintains an ACC of **87.75% (a marginal 2% drop)**, demonstrating its robustness.
>
> We'll add all experiments above in our revision.
>
> **W4**: Consistency Issues.
>
> **R4**: Thanks for pointing them out! For the term 'six inference steps,' our original intention was 'six sampling steps,' while the others are indeed typos. We will revise them and perform a thorough proofreading in our revision. Thank you again for your careful reading!

---

> > ### Author Rebuttal · Reviewer_VdxB · 2026-04-03
> >
> > Thank you for the detailed rebuttal. The response adequately addresses my main concerns by clarifying the empirical nature of the contribution and committing to revise the framing, specifying a held-out threshold calibration protocol with confidence statistics and transfer/ablation results, adding more targeted robustness evaluations, and resolving the presentation inconsistencies I noted. Overall, the rebuttal strengthens the paper and better aligns its claims with the evidence.

---

> > > ### Author Response · Authors · 2026-04-03
> > >
> > > Dear reviewer, thank you so much for your positive feedback! It encourages us a lot! We will ensure that the revised version includes the new experiments and corresponding discussions.

---

### Decision · Program_Chairs · 2026-04-30

**Decision:**

Accept (regular)

**Comment:**

The paper introduces an effective non-intrusive detection method, RECOVER, for identifying unauthorized data usage by leveraging the stronger inversion robustness of fine-tuned images compared with other images. This method enables a more flexible copyright authentication framework without requiring watermark injection or a pre-fine-tuned model. All reviewers recognized the contribution of the proposed framework and its greater practicality compared with previous methods.

However, reviewer VdxB argued that the proposed method is supported more by empirical evidence than by theoretical analysis, particularly regarding inversion robustness for copyright authentication. Reviewers VdxB and yVgX also raised concerns about the completeness of the experimental analysis. In addition, reviewers YCTN and yVgX have the question of the applicability of the method to modern diffusion and flow-matching architectures. Reviewer YCTN further noted that the reliance on white-box or grey-box settings may limit the method’s practicality in real-world scenarios.

In response, the authors promised to improve the presentation, provide additional explanations, and include more experiments on modern diffusion models based on diffusion transformers, such as FLUX.1 and SD3, as well as further robustness experiments under different attacks. The paper ultimately received three consistent weak accept recommendations. Therefore, it is recommended for acceptance after proper polishing in the final manuscript.